# Biochemical and structural characterization of the BioZ enzyme engaged in bacterial biotin synthesis pathway

Sitao Zhang[1,2,10], Yongchang Xu[2,10], Hongxin Guan[1,10], Tao Cui[3,10], Yuling Liao[4,10], Wenhui Wei[2,10], Jun Li[2,5], Bachar H. Hassan[6], Huimin Zhang[2,7], Xu Jia[8], Songying Ouyang [1✉] & Youjun Feng [2,8,9✉]

Biotin is an essential micro-nutrient across the three domains of life. The paradigm earlier step of biotin synthesis denotes "BioC-BioH" pathway in *Escherichia coli*. Here we report that BioZ bypasses the canonical route to begin biotin synthesis. In addition to its origin of *Rhizobiales*, protein phylogeny infers that BioZ is domesticated to gain an atypical role of β-ketoacyl-ACP synthase III. Genetic and biochemical characterization demonstrates that BioZ catalyzes the condensation of glutaryl-CoA (or ACP) with malonyl-ACP to give 5′-keto-pimeloyl ACP. This intermediate proceeds via type II fatty acid synthesis (FAS II) pathway, to initiate the formation of pimeloyl-ACP, a precursor of biotin synthesis. To further explore molecular basis of BioZ activity, we determine the crystal structure of *Agrobacterium tumefaciens* BioZ at 1.99 Å, of which the catalytic triad and the substrate-loading tunnel are functionally defined. In particular, we localize that three residues (S84, R147, and S287) at the distant bottom of the tunnel might neutralize the charge of free C-carboxyl group of the primer glutaryl-CoA. Taken together, this study provides molecular insights into the BioZ biotin synthesis pathway.

[1] Laboratory of Innate Immune Biology of Fujian Province, Provincial University Key Laboratory of Cellular Stress Response and Metabolic Regulation, Biomedical Research Center of South China, Key Laboratory of OptoElectronic Science and Technology for Medicine of the Ministry of Education, College of Life Sciences, Fujian Normal University, Fuzhou, Fujian, China. [2] Department of Pathogen Biology & Microbiology and General Intensive Care Unit of Second Affiliated Hospital, Zhejiang University School of Medicine, Hangzhou, Zhejiang, China. [3] School of Life Sciences, Northwestern Polytechnical University, Xi'an, Shaanxi, China. [4] Guangdong Provincial Key Laboratory of Protein Function and Regulation in Agricultural Organisms, College of Life Sciences, South China Agricultural University, Guangzhou, Guangdong, China. [5] Key Laboratory of Bioorganic Synthesis of Zhejiang Province, College of Biotechnology and Bioengineering, Zhejiang University of Technology, Hangzhou, Zhejiang, China. [6] Stony Brook University, Stony Brook, NY, USA. [7] Carl R. Woese Institute for Genomic Biology, University of Illinois at Urbana-Champaign, Urbana, IL, USA. [8] Non-coding RNA and Drug Discovery Key Laboratory of Sichuan Province, Chengdu Medical College, Chengdu, Sichuan, China. [9] College of Animal Science, Zhejiang University, Hangzhou, Zhejiang, China. [10] These authors contributed equally: Sitao Zhang, Yongchang Xu, Hongxin Guan, Tao Cui, Yuling Liao, Wenhui Wei. ✉email: ouyangsy@fjnu.edu.cn; fengyj@zju.edu.cn

Biotin is an essential enzyme cofactor for all living organisms across the three domains of life[1–3]. The importance of biotin as a prosthetic group is attributed to its participation in the transfer of one carbon units ($CO_2$) in a number of reactions like carboxylation, decarboxylation, and trans-carboxylation[1,3]. Biotin-dependent enzymes include acetyl-CoA carboxylase subunit B (AccB) of fatty acid synthesis[4], methyl-crotonyl-CoA carboxylase (MCC) and propionyl-CoA carboxylase (PCC) of amino acid metabolism[5,6], and pyruvate carboxylase (PC) of gluconeogenesis[7,8]. Unlike plants and a number of microbes capable of de novo synthesis of biotin, mammals like humans rely on exogenous supply of biotin from either diet or the gut symbiotic microbiota[9]. Dysfunction in biotin homeostasis is associated with a number of human pathologies, especially neurological disorders[8,10]. In addition to its classical roles in biotin-dependent enzymes, certain noncanonical biotin utilizations were also reported, which include histone biotinylation and signal transductions in patients[8].

Structurally, biotin is a sulfur-containing derivative of a C7 fatty acid, which consists of a bicyclic ring and a valeric acid side chain[11]. Since its initial discovery in 1901[11–13], biochemical and physiological roles of biotin have been gradually elucidated[14–16]. Our current knowledge on biotin metabolism mostly arisen from studies with the two model organisms, *Escherichia coli* and *Bacillus subtilis*[9,13,16]. However, our understanding of the diversity of bacterial biotin biosynthetic pathways remains fragmentary. In general, biotin synthesis proceeds in two stages: (i) the generation of pimelate moiety, an atypical α, ω-dicarboxylic acid of seven-carbons; and (ii) the assembly of a fused heterocyclic rings of biotin[16–18]. The latter steps are highly conserved in that it consistently proceeds via four successive reactions catalyzed by four unique enzymes[2]. Namely, they correspond to 8-amino-7-oxo-nonanoate synthase (AONS) encoded by *bioF*, 7,8-diaminononanoate synthase (DANS), the product of *bioA* gene, dethiobiotin synthetase (DTBS) by *bioD*, and the biotin synthase BioB[13,16]. In contrast, the initial steps of pimelic acid synthesis differ greatly amongst the varied bacterial lineages[2,19]. In the paradigm organism *E. coli*, the "BioC-BioH" machinery exploits a disguising strategy of "methylation-to-demethylation"[20], which allows methyl-malonyl-ACP thioester as an unusual primer to trigger two cycles of type II fatty acid synthesis (FAS II), giving pimeloyl-ACP[20–22]. Not surprisingly, a large number of *bioH*-negative bacteria that retain *bioC*[2,19], have evolved distinct nonhomologous isoenzymes to compensate for the loss of BioH[23]. As expected, all the BioH functional equivalents belong to the family of α/β hydrolases[23–25], which include BioG of *Haemophilus*[23,26], the *Synechococcus* BioK[23], BioJ in *Francisella*[25,27], and BioV from *Helicobacter*[24]. This increases the diversity of demethylases in the context of biotin synthesis.

The second well-studied mechanism by which the pimeloyl moiety is generated, involves the "BioI-BioW" pathway of *Bacillus subtilis* and its relatives[17,28–30]. Among them, the cytochrome P450 enzyme BioI in *Bacillus*, cleaves a specific carbon–carbon bond (C7–C8) of long chain acyl-ACP to liberate an equivalent of pimelic acid[29,31]. Complex structures of BioI liganded with different acyl-ACPs illustrate how fatty acyl chains are shaped into a "U-turn" above the heme iron cofactor within the enzyme active site[30]. BioW of *Bacillus* acts as a pimeloyl-CoA synthetase[28,32], converting free pimelic acids into the thioester of pimeloyl-CoA via an ATP-dependent mechanism[17]. Crystal structures of BioW enzymes from *B. subtilis*[33] and *Aquifex aeolicus*[34] provided mechanistic insights into the ligation of coenzyme A (CoA) with pimelic acid. $^{13}C$-labeling experiments by Manandhar and Cronan[17] supplemented in vivo evidence that free pimelate also originates from the FAS II pathway in *B. subtilis*. In contrast to pimeloyl-ACP being preferentially used by the *E. coli* BioF as a substrate[13,20,35], pimeloyl-CoA is exclusively utilized by the *B. subtilis* BioF as a *bona fide* precursor of biotin synthesis[18]. Therefore, the "BioI-BioW" pathway of *B. subtilis* represents a unique earlier route committed to pimelate production[30,33], which is far distinct from the "BioC-BioH" route in *E. coli*[20,22].

The machinery of pimelate production seems strikingly different in the phylum of α-proteobacteria (like *A. tumefaciens*) because it has neither "*bioC-bioH*" nor "*bioI-bioW*" homologous pathways[2]. In contrast, a *fabH*-like gene, *bioZ*, seems to be domesticated into a unique *bioBFDAZ* operon in certain α-proteobacterial species (Supplementary Fig. 1)[2]. This promoted the hypothesis that BioZ defines a third pathway for biotin precursor synthesis. However, the biochemical mechanism of BioZ remains an enigma as of this manuscript preparation. In this work, we aim at shedding some light on the role of BioZ in biotin precursor synthesis. Not only do we propose that BioZ arises from the FabH branch within the family of FAS enzymes, but also present structural basis for BioZ action in biotin synthesis. More importantly, we report integrative evidence that BioZ initiates the synthesis of 5-keto-pimeloyl-ACP from glutaryl-CoA along with malonyl-ACP. Therefore, this finding constitutes a functional proof that BioZ bypasses the early canonical steps of biotin synthesis, unveiling a previously unknown for pimelate production.

## Results

**Requirement of biotin for *A. tumefaciens* growth**. The *bioBFDA* operon of *A. tumefaciens* encodes four enzymes responsible for the latter steps of biotin synthesis, assuring the physiological requirement of biotin (Fig. 1a). Presumably, the removal of this operon impairs its capability of biotin synthesis, giving the biotin-auxotrophic strain Δ*bioBFDA* (Fig. 1a). Therefore, it is in rational to establish a DTB/biotin bioassay with the Δ*bioBFDA* mutant as an indicator strain. As expected, the Δ*bioBFDA* mutant cannot appear on the M9 minimal medium without biotin in this biotin assay (Fig. 1b). The growth defect of this mutant was significantly restored by the exogenous addition of biotin (2–10 pmol, Fig. 1b), rather than its precursor DTB (Fig. 1c). The inability of DTB in supporting bacterial growth of Δ*bioBFDA* is due to the lack of BioB, an essential biotin-converting enzyme from DTB (Fig. 1a). On the basis of DTB/biotin assay, cross-feeding experiments were also performed involving two different combinations (donor and recipient). Namely, (i) the wild-type strain of *A. tumefaciens*, NTL4 is a DTB/biotin producer, and in the Δ*bioBFDA* mutant acts as a recipient strain; (ii) cell-free bacterial supernatants (*A. tumefaciens* and *Klebsiella pneumoniae* Kp24 strain) serve as the DTB/biotin donors, whereas the ER90 (Δ*bioFCD*) strain of *E. coli* functions as a DTB/biotin sensor. Not surprisingly, the *A. tumefaciens* NTL4 strain (2 μl of log-phase culture) can cross-feed the biotin-auxotrophic strain of Δ*bioBFDA*, allowing its growth on the nonpermissive condition (Fig. 1d). This indicated that appreciable level of DTB/biotin is excreted by the *A. tumefaciens* NTL4 strain into its growth environment. In fact, we found that the cell-free supernatant of NTL4 strain (~5 μl) supports robust growth of ER90, as biotin (5 pmol) does (Fig. 1e). However, no obvious growth was detected despite of the culture medium (5 μl) of *K. pneumoniae* Kp24 strain. Evidently, this observation confirms that (i) biotin is an essential micronutrient for *A. tumefaciens* growth; and (ii) the NTL4 strain of *A. tumefaciens* can secret more DTB into growth medium than *K. pneumoniae* Kp24 does.

**Functional replacement of BioC-BioH with BioZ**. In the context of lipid metabolism, the phylum of α-proteobacterium such as *A. tumefaciens* has a higher frequency of gene duplication and

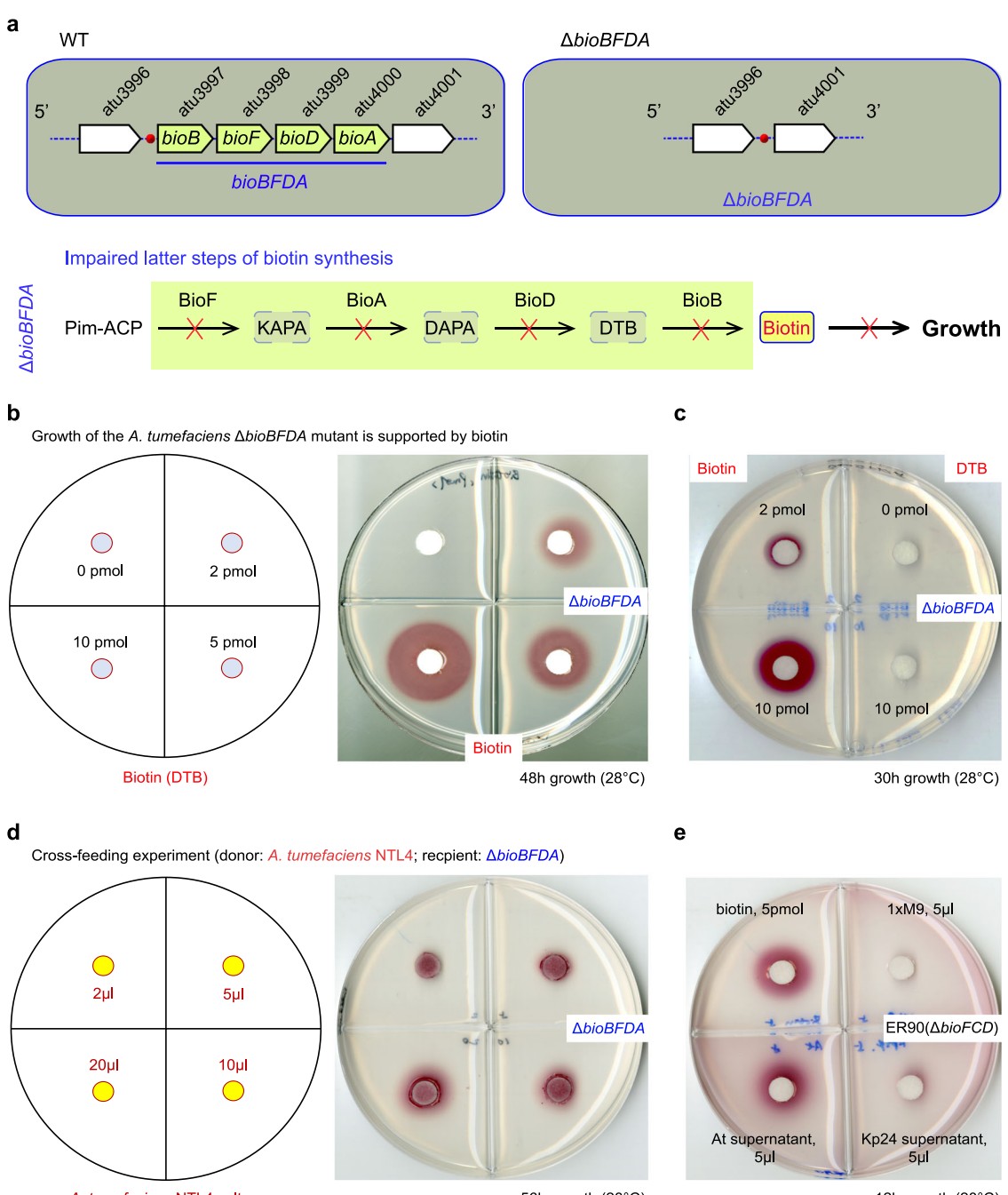

**Fig. 1 Requirement of biotin for *Agrobacterium* growth. a** Schematic illustration of the biotin-auxotrophic strain of *A. tumefaciens* (Δ*bioBFDA*). The removal of biotin operon *bioBFDA* from A. *tumefaciens* NTL4 interrupts the biotin synthesis pathway of *A. tumefaciens*, and therefore interferes bacterial growth on the condition without the supplementation of exogenous biotin. **b** Growth rescue of the Δ*bioBFDA* mutant by the addition of exogenous biotin. **c** Unlike biotin, its precursor dethiobiotin (DTB) fails to allow the Δ*bioBFDA* mutant to appear on the nonpermissive growth condition without any biotin. **d** The growth of the Δ*bioBFDA* mutant on the biotin-lacking medium is restored by the cross-feeding with the wild-type strain of *A. tumefaciens* NTL4. **e** The cell-free growth culture (i.e., supernatant) of *A. tumefaciens* NTL4, (rather than the negative control, *Klebsiella pneumoniae* strain 24) cross-feeds the biotin auxotroph ER90 (Δ*bioFCD*) of *E. coli*. Here, the *A. tumefaciens* NTL4 was pelleted and then suspended with 1× PBS. It indicated that *A. tumefaciens* NTL4 secrets biotin (mostly its DTB precursor) into the growth medium/environment[39]. The strain FYJ283 (Δ*bioBFDA*)[39] appeared as an indicator strain in the biotin (DTB) bioassay (**b**–**c**), and also acted as a recipient strain in the cross-feeding experiment of *A. tumefaciens* NTL4 (**d**). The biotin/DTB bioassay was routinely performed as earlier described[20,39]. At *Agrobacterium tumefaciens*, Kp24 *Klebsiella pneumoniae* strain 24.

amplification when compared to *E. coli* (Supplementary Fig. 1)[2]. On the two chromosomes of *A. tumefaciens*, an array of redundant *fab*-related loci is dispersed (Supplementary Fig. 1), which includes two *fabI* genes [*fabI1* (*atu0149*) and *fabI2* (*atu0752*)], 4 *fabF* genes [*fabF1* (*atu1097*) to *fabF4* (*atu4216*)], and 4 *acpP* genes [*acpP1* (*atu1096*) to *acpP4* (*atu8162*)]. A similar distribution is seen in

both *Brucella melitensis*[36] and *Rhizobium* sp. IRBG74[37,38]. While *fabG* is present as a single copy in *A. tumefaciens*, it appears as five copies in *B. melitensis* and seven copies in *Rhizobium* sp. IRBG74 (Supplementary Fig. 1). Intriguingly, a second copy of *fabH* gene that presumably encodes a β-ketoacyl-ACP synthase III (KAS III) of fatty acid synthesis, is consistently integrated into a single

*bioBFDAZ* operon of biotin synthesis in certain α-proteobacterial species, like *A. tumefaciens* (Supplementary Figs. 1-2)[2]. Therefore, it was called *bioZ* (*fabH2*), in *A. tumefaciens*[39], *B. melitensis*[36], and *Rhizobium*[37,38] (Supplementary Fig. 1). An earlier genetic study by Sullivan and coworkers[38], demonstrated that a transposon-based inactivation of *bioZ* impairs the growth of *Mesorhizobium* sp. strain R7A on media lacking biotin and the complementation of the *bioZ* gene into the *bioH* mutant of *E. coli* restored its viability on biotin-lacking media.

To functionally characterize BioZ in biotin metabolism, we cloned *bioZ* from three different species into the arabinose-inducible plasmid pBAD24 (Supplementary Table 1). We then assayed the in vivo role of BioZ using *E. coli* biotin-auxotrophic strains (Δ*bioH*, Δ*bioC*, or the Δ*bioH*/Δ*bioC* double mutant) (Supplementary Figs. 2-5a). As predicted, the introduction of *Brucella bioZ* can rescue bacterial growth of both Δ*bioH* and Δ*bioC* on the biotin-free, nonpermissive growth condition (Supplementary Fig. 3). Furthermore, the expression of *bioZ* confers efficient growth of the Δ*bioH*/Δ*bioC* double mutant on biotin-deficient media (Supplementary Fig. 3). Similar results were obtained with the *A. tumefaciens* and *Rhizobium bioZ* (Supplementary Fig. 4). Consequently, BioZ (FabH2) bypasses the roles of "BioC-BioH" in the canonical earlier steps of biotin precursor synthesis. From an evolutionary point of view, this result suggested that BioZ has been domesticated from a prototypical fatty acid synthetic enzyme to gain an additional role in biotin synthesis. Thus, BioZ could invoke a third pathway by which the biotin precursor pimelate is synthesized in α-proteobacterium, esp. *A. tumefaciens*.

**Origin and phylogeny of BioZ.** To probe the possible origin of BioZ, a database-wide BLAST search against GenBank was performed with manual collation. The majority of protein candidates obtained belong to the family of KAS enzymes: KAS I (FabF), KAS II (FabB), and KAS III (FabH1 and FabH2). BLAST search resulted in almost 150 hits, including 41 FabF, 15 FabB, 50 FabH, and 43 BioZ (Fig. 1a). Since these candidates were highly variable by homology searches, we thereby revaluated their homologous domains via Pfam server (https://pfam.xfam.org/). Unlike the *fab* loci scattered on chromosomes, the *fabH*-like locus *bioZ* is frequently found to localize within the biotin synthesis gene cluster (Supplementary Fig. 1). These results, taken into the genetic context of neighboring genes, suggested that *bioZ* probably arises from recent integration events.

As shown in the unrooted tree, two distinct clades (FabF and FabH) are present (Fig. 2a). The FabH clade seem to be classified into three subgroups, one of which belongs to BioZ (Fig. 2a). Evidently, BioZ has likely evolved from FabH in certain species of *Rhizobiales*. During the domestication process with unknown selection pressure, such FabH ancestors might be accidently harnessed or hijacked to participate in biotin synthesis, despite its relative-low initial efficiency. Likewise, FabB is represented as a subclan within the FabF group, far different from the former FabH lineage (Fig. 2a). This suggests that FabF can be an ancestor of FabB. Noteworthy, *A. tumefaciens* C58 has evolved all the homologs across the aforementioned subclades (Fig. 2a), the specialized roles of which might benefit the evolutionary adaptability and metabolic fitness of *Agrobacterium*.

The phylogeny of BioZ homologs further inferred their potential evolutionary correlation across different species (Fig. 2b). Intriguingly, three highly similar *bioZ* genes (mll9094, mll5827, and mlr6070) were found in *Mesorhizobium japonicum* MAFF303099, indicating that the origination and evolution of BioZ is connected with gene duplication or horizontal transfer (Fig. 2b). In brief, the pMLa plasmid-borne mll9094 is 91.74% similar to the chromosomal

mll5827 at amino acid sequence level. And mlr6070 is 86.52% identical to mll5827, of which sequence is 118 aa in size, significantly shorter than the BioZ prototype (>300aa). However, the core domains of this truncated chimera and mll5827 are both indicatives of BioZ-type enzymes. In particular, we observed that the loci adjacent to *mlr6070* (i.e., *mll6064*, *mll6066-mll6068*, *mlr6069*, and *mll6075*) encode a number of transposal elements[40,41]. This underscored the possibility of on-going dissemination/variation of *bioZ* gene. The truncated version of *mlr6070* is the remnant or degenerative in the domestication/evolution of *bioZ*, or alternatively, this homologous gene might provide fragment insertion sites. Therefore, these observations supported our proposal that *bioZ* originates from *Rhizobiales* (Fig. 2b). In summary, *bioZ* could be subjected to independent diversification rather than co-evolution with other biotin synthesis related genes.

**Biochemical insights into BioZ action.** To address its biochemical activity, we expressed BioZ from different organisms using a prokaryotic expression system (Supplementary Fig. 5a). Among the three *bioZ*-inserted constructs we examined, only AtBioZ could be overexpressed in the form of partially soluble protein, exhibiting an apparent mass of ~36 kDa (Supplementary Fig. 5b). The purity of the recombinant AtBioZ protein was determined with gradient PAGE (Supplementary Fig. 5b). Then, mass spectrometry validated its polypeptide fingerprint with a 44% sequence coverage (Supplementary Fig. 5c). Both gel filtration and chemical cross-linking experiment of AtBioZ revealed that its solution structure is of dimeric stoichiometry (Supplementary Fig. 5b, d). Despite of its capability to complement the *E. coli* Δ*bioH* mutant (Supplementary Figs. 3-4), AtBioZ cannot liberate the methyl group from the paradigm substrate of BioH, methyl-pimeloyl-ACP, to give pimeloyl-ACP, in the in vitro enzymatic system (Supplementary Fig. 6a-b). Using the in vitro reconstituted system of DTB/biotin synthesis, the resultant product by BioJ restored the growth of the biotin-auxotrophic ER90 strain on the biotin-lacking, nonpermissive condition[27,42], whereas not for AtBioZ (Supplementary Fig. 6c-d). This underscored that BioZ represents a previously-unrecognized mechanism, and needed further experimental determination.

Given that AtBioZ originated from FabH, we then asked the two questions: (i) whether or not it retains the β-ketoacyl-ACP synthase III activity; (ii) how the primer substrate specificity differs between AtBioZ (AtFabH2) and AtFabH1. Thus, we purified the enzymes of fatty acid synthesis and reconstituted the in vitro FAS II system (Supplementary Figs. 7 and 8a). As for the initial reaction of fatty acid synthesis, a variety of FabH enzymes might recognize acyl-CoAs of different carbon chain lengths (Supplementary Fig. 8). Of note, the condensation of primer substrate by FabH with malonyl-ACP proceeds via the successive activities of FabG, FabZ (FabA) and FabI using NADH and NADPH as coenzymes. As expected, the *E. coli* FabH enzyme (EcFabH, control) was observed to ligate exclusively acetyl-CoA with malonyl-ACP to form aceto-acetyl-ACP which enters the FAS II cycle, giving butyryl-ACP (C4-ACP) (Fig. 3a and Supplementary Fig. 8b). The AtFabH1 was detected active with at least two substrates, acetyl-CoA (C2-CoA) and butyryl-CoA (C4-CoA), suggesting its enzymatic promiscuity (Supplementary Fig. 8c-d). Unlike the scenario with AtFabH1, it seemed likely that AtBioZ catalyzes the condensation of primer substrate glutaryl-CoA with malonyl-ACP, producing a pool of C7-ACP (Supplementary Figs. 8e and 9). The resultant product mixture displayed a similar mobility of pimeloyl-ACP (positive control), slightly above holo-ACP and malonyl-ACP in the separation by conformationally sensitive, 0.5 M urea/PAGE (17.5%, pH9.5) (Fig. 3a). The pimeloyl-ACP positive control used in our assay

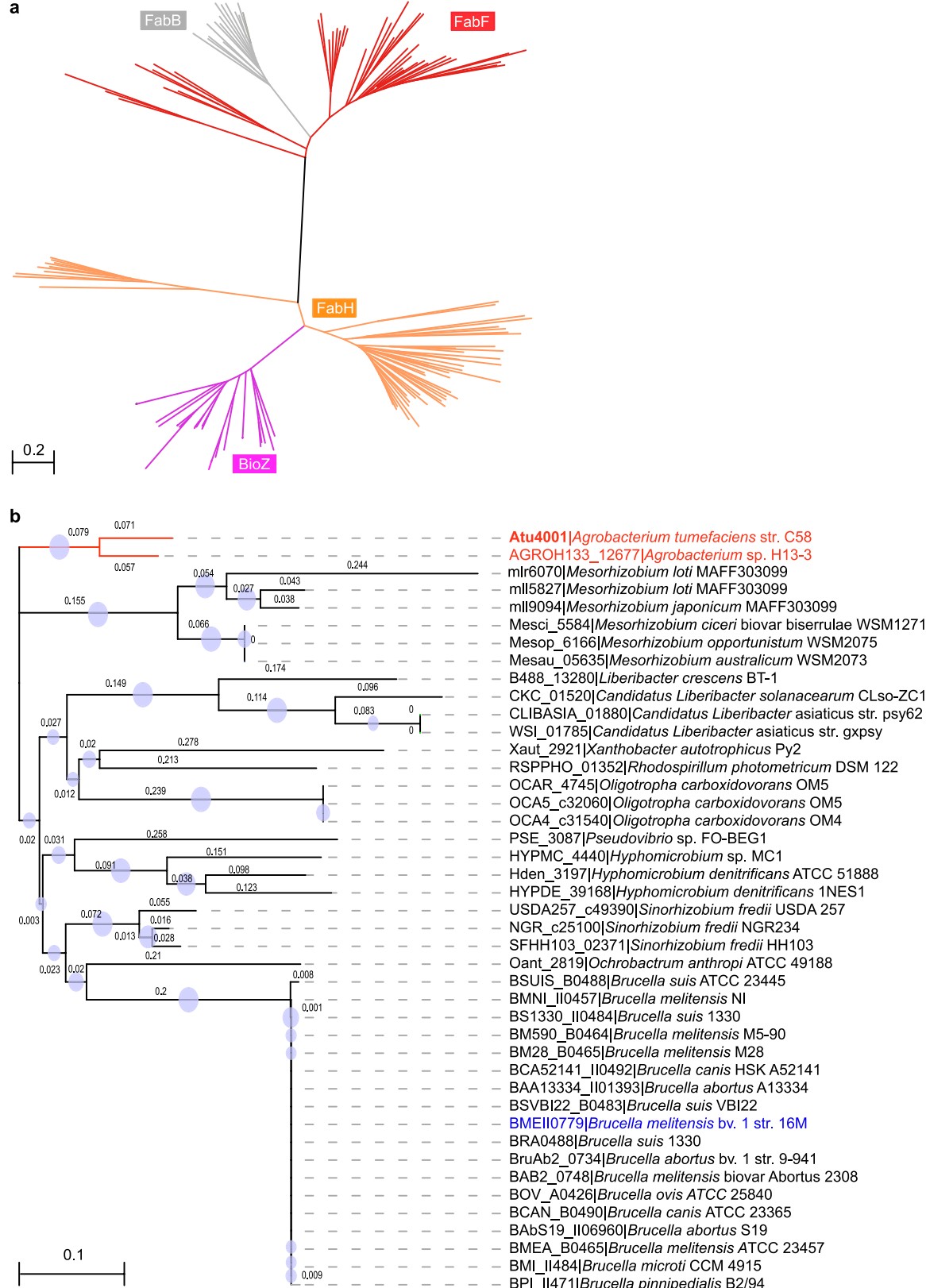

**Fig. 2 Phylogeny of BioZ and its paralogs. a** An unrooted tree of BioZ proteins and its putative homologs (FabH, FabB, and FabF). The phylogenetic tree was generated with the MEGA7 software using the NJ method (bootstrap: 1000 replicates). Two major clades of KAS enzymes are shown: FabF in red and FabH in orange. In contrast, FabB (shown in gray) is phylogenetically positioned as a sub-branch of FabF clade and BioZ was localized to a sub-branch of FabH clade. Therefore, we speculated that FabB originates from FabF (KAS II), and a recent ancestor of BioZ arises from FabH, the KAS III enzyme within the KAS pan-family. **b** Phylogenetic relationships of BioZ paralogs. Phylogeny was constructed using the MEGA7 software involving the NJ method (bootstrap: 1000 replicates, bootstrap values indicated by circle sizes). BioZ of *Agrobacterium* and *Brucella melitensis* are colored in red and blue, respectively.

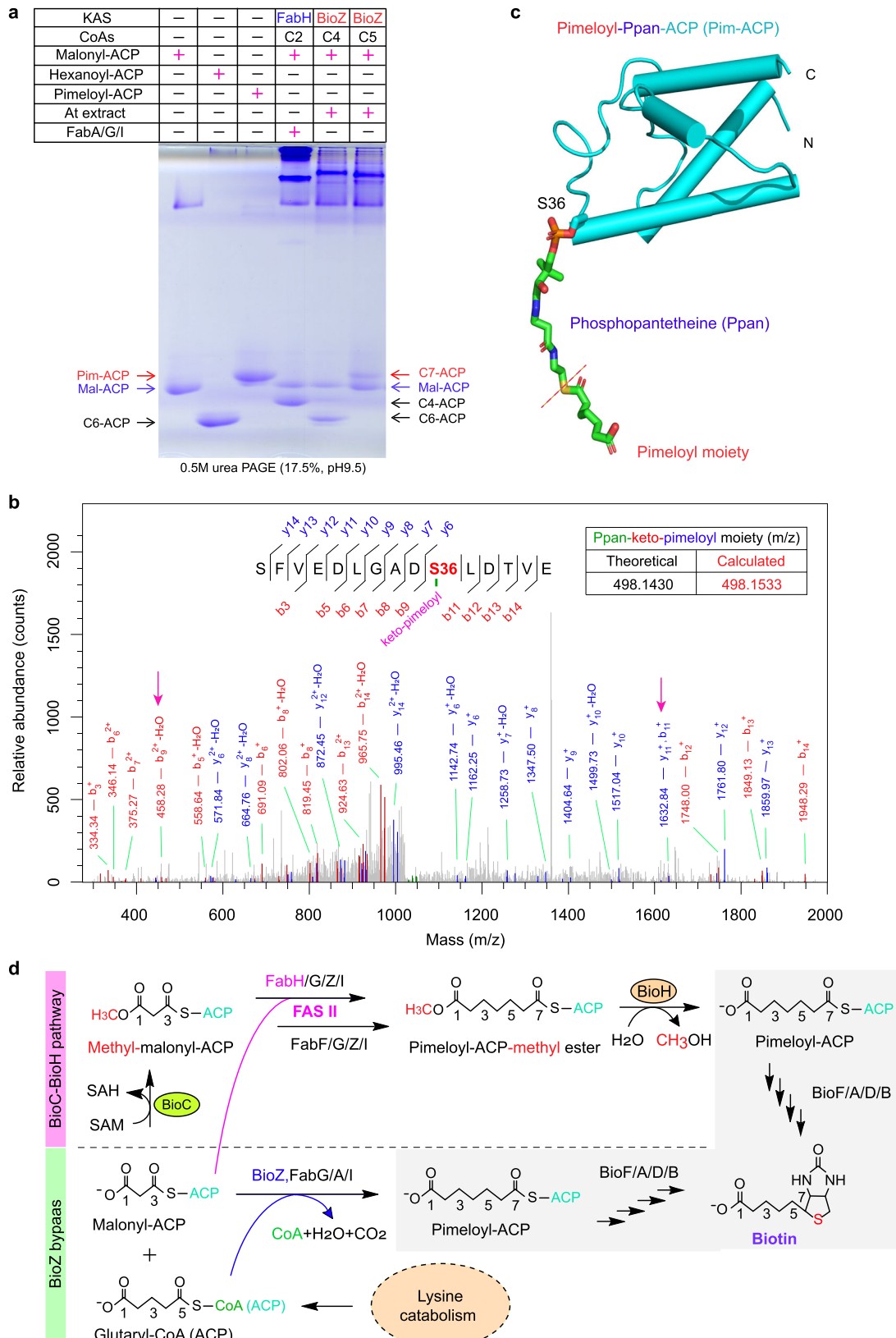

Figure a: table and gel

| KAS | – | – | – | FabH | BioZ | BioZ |
|---|---|---|---|---|---|---|
| CoAs | – | – | – | C2 | C4 | C5 |
| Malonyl-ACP | + | – | – | + | + | + |
| Hexanoyl-ACP | – | + | – | – | – | – |
| Pimeloyl-ACP | – | – | + | – | – | – |
| At extract | – | – | – | – | + | + |
| FabA/G/I | – | – | – | + | – | – |

0.5M urea PAGE (17.5%, pH9.5)

Pim-ACP →
Mal-ACP →
C6-ACP →
← C7-ACP
← Mal-ACP
← C4-ACP
← C6-ACP

**c** Pimeloyl-Ppan-ACP (Pim-ACP)

S36
Phosphopantetheine (Ppan)
Pimeloyl moiety

**b**

| Ppan-keto-pimeloyl moiety (m/z) | |
|---|---|
| Theoretical | Calculated |
| 498.1430 | 498.1533 |

S F V E D L G A D S36 L D T V E

**d**

BioC-BioH pathway

Methyl-malonyl-ACP →(FabH/G/Z/I, FAS II, FabF/G/Z/I)→ Pimeloyl-ACP-methyl ester →(BioH, H₂O, CH₃OH)→ Pimeloyl-ACP →(BioF/A/D/B)→ Biotin

BioC: SAH ← SAM

BioZ bypaas

Malonyl-ACP + Glutaryl-CoA (ACP) →(BioZ, FabG/A/I; CoA+H₂O+CO₂)→ Pimeloyl-ACP →(BioF/A/D/B)→ Biotin

Lysine catabolism → Glutaryl-CoA (ACP)

was produced in our lab by the demethylation of pimeloyl-ACP methyl ester using the BioJ enzyme (Supplementary Fig. 7b). Not surprisingly, we also observed that BioZ possess the activity on glutaryl-ACP along with malonyl-ACP, giving a C7-ACP product (Supplementary Fig. 10a-b). In fact, our result of isothermal titration calorimetry (ITC) verified efficient binding of BioZ to glutaryl-ACP with the stoichiometry of N = 0.978 ± 0.028 and Kd = 6.167 ± 0.068 μM (Supplementary Fig. 11). This might be

**Fig. 3 A role of BioZ in the FAS II-involving biotin biosynthesis. a** In vitro biosynthesis of pimeloyl-ACP and/or its precursors. Using the FAS II system, BioZ catalyzes the synthesis reaction of pimeloyl-ACP from malonyl-ACP and glutaryl-CoA (ACP). The reaction mixture was separated with conformation-sensitive urea polyacrylamide gel electrophoresis (PAGE). A representative result is given from three trials. Of note: 17.5% PAGE (pH 9.5) containing 0.5 M urea was used here. The two controls (C4-ACP and C6-ACP) served as standards/markers for this conformation-sensitive urea gel. C7-ACP denotes four species of ACP attached with an acyl seven-carbon fatty acyl chain, namely, 5-keto-pimeloyl-ACP, 5-hydroxyl-pimeloyl-ACP, enoyl-pimeloyl-ACP, and pimeloyl-ACP. **b** MS/MS identification of 5-keto-pimeloyl-ACP, a primary product from the BioZ reaction coupled with FAS II, using glutaryl-CoA and malonyl-ACP as substrates. The use of MS/MS allowed us to detect the presence of four C7-ACP species in the above reaction system. As for an initial product of BioZ reaction, 5-keto-pimeloyl ACP, a 15-residue peptide fragment of interest is given. The C7-fatty acyl modification with high reliability is localized on the conserved Serine 36 of ACP. The two peaks of peptide fragments indicated with pink arrows were used to determine C7 acyl modification. The resultant mass was 498.1533, which is close to the theoretical mass (498.143) of Ppan-linked keto-pimeloyl moiety. Of note, it might be not as stable as pimelic acid. **c** Cartoon illustration of the pimeloyl-ACP structure. This was generated from the complex structure of methyl-pimeloyl-ACP and BioH (PDB: 4ETW) with appropriate modifications. **d** A scheme for BioZ bypassing the canonical early steps of biotin synthesis. Unlike the paradigm "BioC-BioH" mechanism of biotin synthesis (above the dashed line), the BioZ reaction bypasses the earlier steps of "BioC-BioH" in biotin biosynthesis (below the dashed line). Recently, glutaryl-CoA is determined to physiologically arise from lysine catabolism in *Agrobacterium* species[47]. Designations: C4-ACP butanoyl-ACP, C6-ACP hexanoyl-ACP, Mal-ACP malonyl-ACP, Glu-ACP glutaryl-ACP, Pim-ACP pimeloyl-ACP.

because that the acyl derivatives of CoA and ACP usually mimic one another in the context of FAS II system. In addition, the AtBioZ was found to retain the activity on butyryl-CoA (C4-CoA), which is generally consistently with that of AtFabH1, but not EcFabH (Fig. 3a and Supplementary Fig. 8b-d).

Subsequently, high-resolution MS/MS was applied to further confirm the generation of C7-ACP (such as pimeloyl-ACP) product from BioZ-catalyzed reaction coupled with the in vitro reconstituted pathway of fatty acid synthesis by FabGAI (and/or cell-free crude extract). First, a number of short C7-fatty acyl modified peptides of ACP was detected in the BioZ/At extract system. Among them, a fragment of 15aa long was calculated by MS/MS to possess the mass addition of 498.1533 with high reliability, which is almost identical to the theoretical mass (498.1430) of phosphopantetheine (Ppan) attached with keto-pimeloyl moiety (Fig. 3b). This modification of 5-keto-pimeloyl moiety was further localized to the Serine 36 (S36) of holo-ACP (Fig. 3b). In particular, MS/MS data retuned several ACP peptides having a pimeloyl modification on the S36 residue (Fig. 3b and Supplementary Fig. 10c). Likewise, using the BioZ/FabGAI system, we were fortunate to track an array of ACP peptides on the S36 site of which three types of different C7-fatty acyl modifications occur (Supplementary Figs. 12-14). Namely, they included 5-keto-pimeloyl moiety (idea mass vs actual mass: 498.143 vs 498.89 in Supplementary Fig. 12), 5-hydroxyl-pimeloyl moiety (idea mass vs actual mass: 500.159 vs 500.147 in Supplementary Fig. 13), and enoyl-pimeloyl moiety (idea mass vs actual mass: 482.1488 vs 482.576 in Supplementary Fig. 14). In principle, these three C7 intermediates are not as stable as its final product pimelic acid. These data proved that AtBioZ catalyzes the condensation of the primer substrate glutaryl-CoA (perhaps ACP) with malonyl-ACP to form 5-keto-pimeloyl-ACP. Upon being coupled FAS II pathway, this unstable intermediate can be converted into pimeloyl-ACP product, i.e., the first precursor for the latter steps (BioF/A/D/B) of biotin synthesis (Supplementary Fig. 9). Thus, we proposed a working model that the α-proteobacterial BioZ enzyme behaves as an atypical member of β-ketoacyl-ACP synthase III to synthesize C7-fatty acyl moiety, bypassing the canonical 'BioC-BioH' earlier steps of pimelate generation (Fig. 3d).

**Structural characterization of BioZ.** Using X-ray crystallography, we crystallized and solved the structure of AtBioZ at a 1.99 Å resolution (Table 1 and Supplementary Fig. 15). The dimeric nature of the AtBioZ protein belongs to the space group of $P\,2_1\,2_1\,2_1$ and contains two protomers in per asymmetric unit (ASU) (Supplementary Fig. 16a), of which the refinement statistics are listed in Table 1. AtBioZ shares a 33% identity with

| Table 1 X-ray crystallography data collection and refinement statistics. | |
| --- | --- |
| **Dataset** | **AtBioZ (PDB: 6KUE)** |
| **Data collection** | |
| Beamline | BL17U1, SSRF |
| Wavelength (Å) | 0.9792 |
| Resolution range (Å)[a] | 55.26–1.99 (2.10–1.99) |
| Space group | $P\,2_1\,2_1\,2_1$ |
| Cell dimensions | |
| a, b, c (Å) | 63.81, 85.78, 110.48 |
| α, β, γ (°) | 90.00, 90.00, 90.00 |
| Total reflections | 1,059,122 (151,343) |
| Unique reflections | 42,056 (6018) |
| Multiplicity | 25.2 |
| Completeness (%) | 99.7 (99.8) |
| *Mean I/sigma(I)* | 15.8/8.9 |
| Wilson B-factor (Å$^2$) | 16.13 |
| *R*-merge | 0.371 (1.114) |
| *R*-means | 0.387 (1.160) |
| *R*-pim | 0.108 (0.322) |
| CC1/2 | 0.995 |
| **Refinement** | |
| Reflections used in refinement | 41942 (4098) |
| Reflections used for R-free | 2038 (195) |
| R$_{work}$ | 0.181 (0.203) |
| R$_{free}$ | 0.229 (0.273) |
| Number of nonhydrogen atoms | 5187 |
| Macromolecules | 4764 |
| Solvent | 423 |
| Protein residues | 654 |
| RMS (bonds) | 0.008 |
| RMS (angles) | 0.96 |
| Average B-factor (Å$^2$) | 20.00 |
| Macromolecules | 19.96 |
| Solvent | 31.28 |
| Ramachandran favored (%) | 98.15 |
| Ramachandran allowed (%) | 1.85 |
| Ramachandran outliers (%) | 0 |

[a]Highest resolution shell is shown in parentheses.

EcFabH[43], while 21% and 18% with EcFabB[44] and EcFabF[45], respectively. Even though the sequence similarity is low amongst these KAS synthases, they seemed to share a common architecture comprising two distinct domains: a lid and a core domain (Fig. 4). The lid domain of AtBioZ contains four α-helices and two β-sheets, which is covering on the 'α/β/α/β/α' core domain (Supplementary Fig. 15a). The core domain of AtBioZ features a

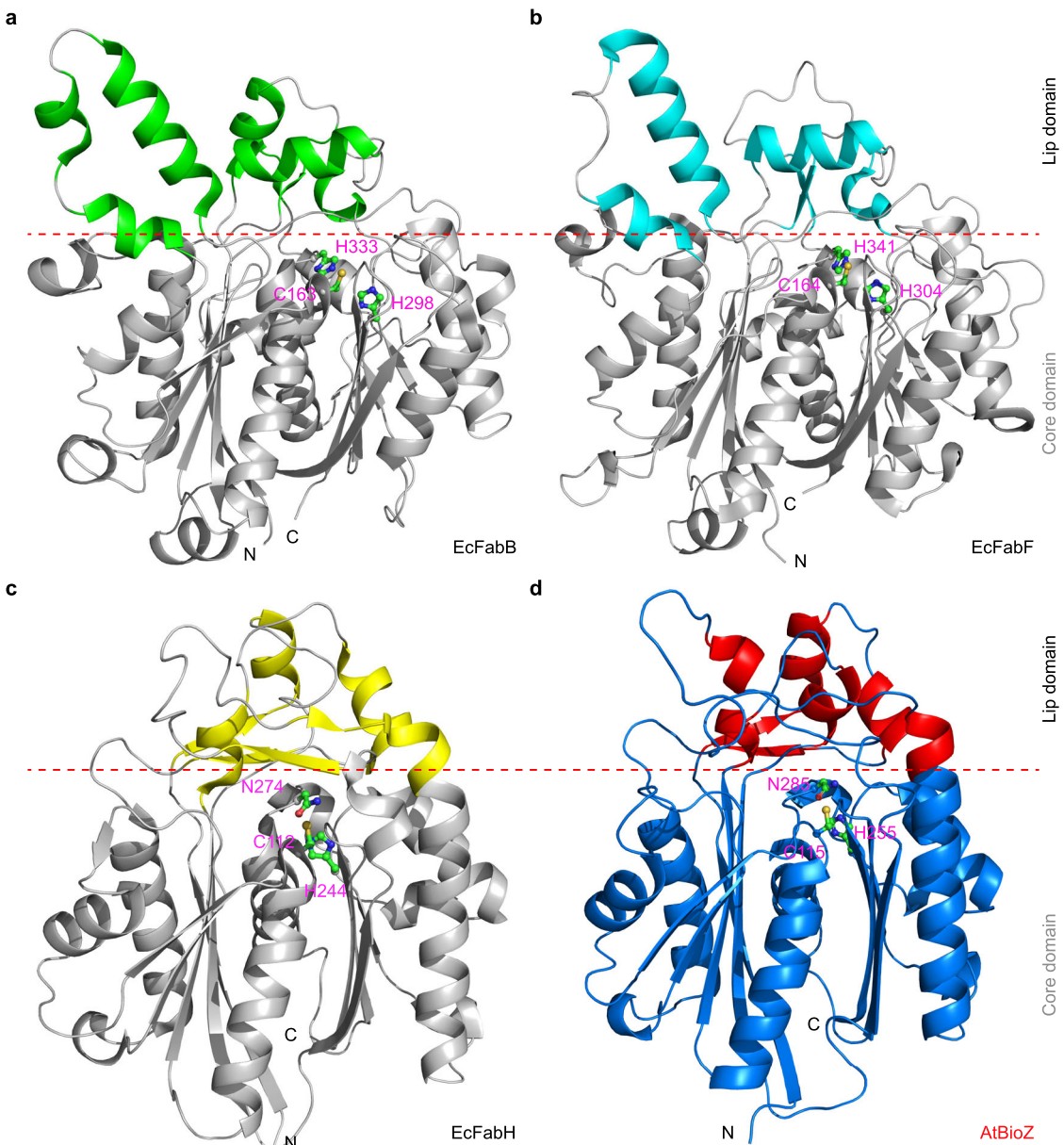

**Fig. 4 Structural comparison of four FAS-type enzymes. a** Ribbon representation of the *E. coli* FabB structure (PDB: 1G5X). **b** Ribbon structure of the *E. coli* FabF (PDB: 2GFW). **c** Ribbon representation of the *E. coli* FabH structure (PDB: 1HN9). **d** Ribbon illustration of the *A. tumefaciens* BioZ structure (PDB: 6KUE). The lid domains in EcFabB (**a**), EcFabF (**b**), and EcFabH (**c**) are separately colored in green, cyan, and yellow, respectively. Whereas, the core domains are colored in silver–gray. As for AtBioZ, the lid domain is shown in red, and its core domain appears in blue (**d**). The three catalytic triad residues are indicated with pink letters.

triple decked sandwich that is formed by nine β-sheets and ten α-helices. In brief, the β-sheets of β1 and β4–β7 are sandwiched between the helices α3–α5 and α6, together with helices of α13–α14 to form the first layer, while the sheets of β8–β11 are sandwiched between the helices of α9–α12 and α6, together with the helix of α13–α14 to form the other layer of the "sandwich" (Supplementary Fig. 15a-b). Interestingly, a putative catalytic triad (C115, H255, and N285, Supplementary Figs. 15 and 19c-d) is centered in the core domain of AtBioZ (Figs. 4d, 5d).

Structural alignment revealed that the root-mean-square deviation (rmsd) of Cα atoms of AtBioZ is 2.8, 2.7, and 1.2 Å, in comparison with FabB, FabF, and FabH, respectively (Fig. 4a, d), while the rmsd between FabB and FabF is 1.4 Å (Fig. 4a, b). Consistent with the evolutionary placement seen in our phylogenic studies (Fig. 2a), the structure of AtBioZ (PDB:

6KUE) is similar to FabH (PDB: 1HN9), whereas FabB (PDB: 1G5X) structurally is more related to that of FabF (PDB: 2GFW) (Fig. 4a, d). Even though the core domain and the active site are highly conserved (Figs. 4 and 5a, d), the lid domains differ dramatically across the four different members of the KAS family enzymes (Fig. 4a, d). In brief, the lid domain of FabB is mainly composed of five α-helices and two β-sheets, and the counterpart of FabF is formed by four α-helices and two β-sheets (Fig. 4a, b). However, the lid domain of FabH/BioZ is composed of two parts: one part is the long loop between β8 and α9 from the first layer of the "sandwich", and the other part contains two regions (β1–α3 and β6–β7) on the second layer of the "sandwich" (Fig. 4c, d and Supplementary Fig. 15). Taken together, our results provide possible explanation for the diversity in substrate specificity/recognition amongst the KAS family enzymes.

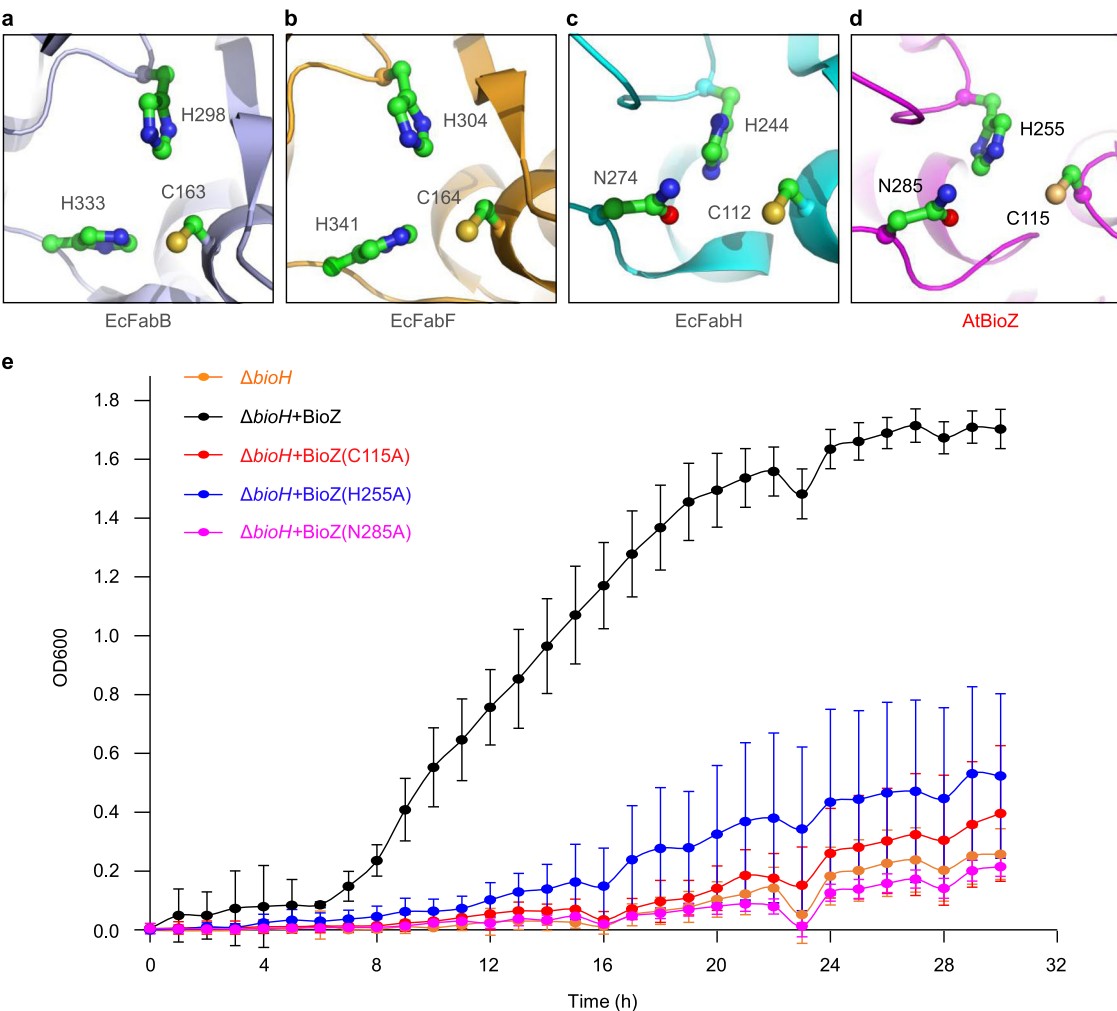

**Fig. 5 Structural and functional analyses of the BioZ catalytic triad. a** Structural snapshot of the catalytic triad (C163, H298, and H333) of EcFabB. **b** Structural analysis of the catalytic triad (C164, H304, and H341) of EcFabF. **c** Structural presentation of the catalytic triad (C112, H244, and N274) of EcFabH. **d** Structural illustration of the catalytic triad (C115, H255, and N285) of AtBioZ. The figures of structures were generated using the PyMol software. **e** Functional assays of BioZ and its catalytic triad mutants using the ΔbioH biotin-auxotrophic strain. The tested strains were grown on nonpermissive M9 minimal media lacking biotin. Growth curves were generated from three independent experiments and displayed in an average ± standard deviation (SD).

**Catalytic triad of BioZ**. FAS enzymes possess an evolutionarily-conserved catalytic center. The catalytic triad (C163, H298, and H333) of EcFabB is almost structurally identical to that of EcFabF (C164, H304, and H341) (Fig. 5a, b). In contrast, the active site residues of AtBioZ correspond to C115, H255, and N285, which are highly similar to those of EcFabH (C112, H244, and N274) (Fig. 5c, d). To test the function of the putative catalytic center residues of AtBioZ, we performed site-directed mutagenesis (C115A, H255A, and N285A) followed by complementation assays on the nonpermissive growth condition without biotin. Experimentally, the ΔbioH biotin-auxotrophic strain was transformed with the wild-type or mutant bioZ plasmid and assayed for growth in liquid M9 minimal media lacking biotin (Fig. 5e). As predicted, the ΔbioH strain grew robustly when complemented with the wild-type bioZ . Whereas, the strains complemented with (C115A or N285A) mutants lost their abilities to grow as illustrated in the growth curves (Fig. 5e). Of note, the H255A mutant retains low activity to allow for some growth (Fig. 5e). This result was also consistent with scenarios seen on the M9 minimal media agar plates. In conclusion, this functionally defines a catalytic triad (C115, H255, and N285) of BioZ,

and highlights varying roles of the aforementioned three residues in the catalytic activity.

**Interaction between malonyl-ACP and BioZ**. Since that BioZ was biochemically confirmed active on malonyl-ACP and glutaryl-CoA (ACP), we then asked the question how this BioZ enzyme interacts with the above two substrates. Although that our earlier efforts failed to co-crystallize AtBioZ complexed with malonyl-ACP or glutaryl-ACP(CoA), the recent availability of crystal structure of FabB crosslinked with ACP[46], allowed us to probe the interplay between malonyl-ACP (or its ACP moiety) and BioZ. Thus, we used the structure of FabB-ACP (2: 1) complex to guide the building of a BioZ-ACP model (Fig. 6a, d). In contrast to the substrate-loading channel of FabB (PDB: 5KOF) that features a closed bottom, the substrate-binding pocket of AtBioZ is open on both ends (Supplementary Fig. 17a, d). The diameter of substrate entrance of both AtBioZ and FabH is measured to be around 5 Å (Supplementary Fig. 17a, c). The depth of the substrate-loading tunnel in AtBioZ is 20 Å, quite longer than the counterpart (15 Å) of FabB (Supplementary Fig. 17b, d).

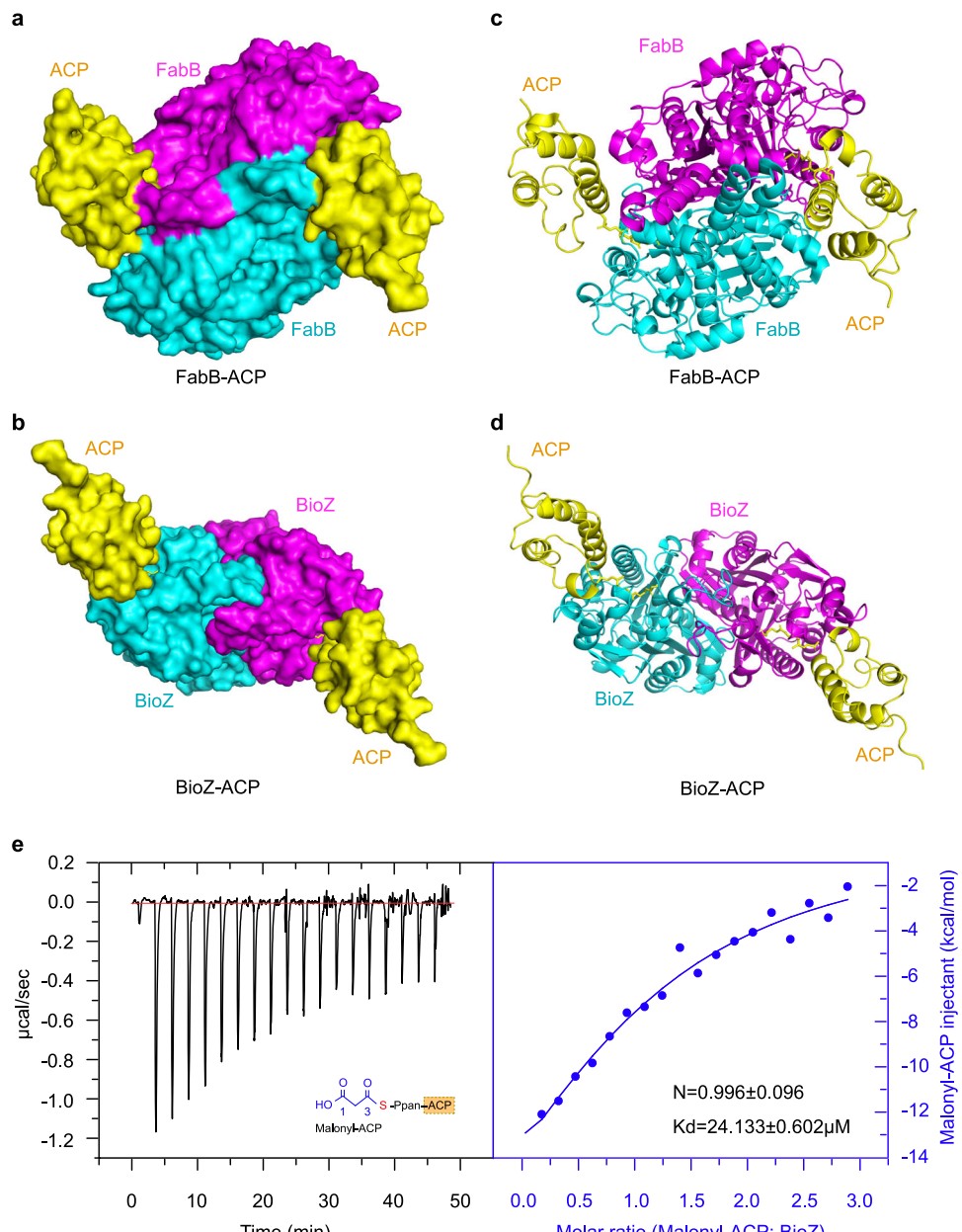

**Fig. 6 Structural and functional analysis of BioZ binding to malonyl-ACP.** Surface structure (**a**) and ribbon illustration (**c**) of the FabB dimer crosslinked with ACP. It was generated with PyMol using crystal structure of FabB-ACP (PDB: 5KOF). Surface structure (**b**) and ribbon illustration (**d**) of the dimeric BioZ complexed with malonyl-ACP. **e** Use of isothermal titration calorimetry (ITC) to probe the interaction between malonyl-ACP and BioZ. A representative result of ITC is shown, and the resultant stoichiometry values (N and Kd) from three independent experiments are given in an average ± SD. The putative binding mode of BioZ to its substrate malonyl-ACP (and/or glutaryl-ACP) was generated through structural superposition. The stoichiometry (malonyl-ACP: BioZ) is 1:1, in that the molar ratio was measured to be ~0.996. The parameter of binding affinity (Kd) is 24.133 ± 0.602 µM.

Unlike that the FabB substrate-binding pocket whose entrance is close to the dimeric interface (Fig. 6a, c), the entrance of the AtBioZ substrate-binding pocket is relatively far away from the dimeric interface (Fig. 6b, d). In the case of FabB-ACP (Supplementary Fig. 19a-b), the ACP protein mainly interacts with one monomer while the acyl group on ACP inserts into the substrate-binding pocket of another monomer (Fig. 6a, c). In other words, both monomers of the FabB dimer cooperate for ACP protein and acyl group binding. The ClusPro-based molecular docking identified a putative interface between BioZ with its cognate ACP (Supplementary Figs. 18a and 19c, d). In contrast to its relatively weak affinity with ACP alone having the stoichiometry (N = 1.00 ± 0.17, Kd = 0.33 ± 0.25 mM, in Supplementary

Fig. 18b), the mutant enzyme of BioZ(C115A) was found to bind modestly malonyl-ACP in our ITC experiments (N = 0.996 ± 0.096, Kd = 24.133 ± 0.602 µM, in Fig. 6e). Therefore, we proposed a "one-to-one" model for BioZ-ACP interaction in which an ACP protein can bind to only one BioZ monomer and the malonyl group of ACP interacts with the same BioZ monomer (Fig. 6b, d). Presumably, FabB-ACP interface mainly relies on an electrostatic interaction between negatively-charged residues on ACP and positively-charged residues on FabB (Fig. 7a)[46]. On the bottom of ACP helix II, Asp35, and Asp38 interact with Arg62, Lys63, and Arg66 on FabB, while Glu47 at the top of helix II interacts with Arg124 and Lys127 on FabB (Fig. 7b)[46]. Similarly, the surface around the entrance of BioZ pocket is enriched with

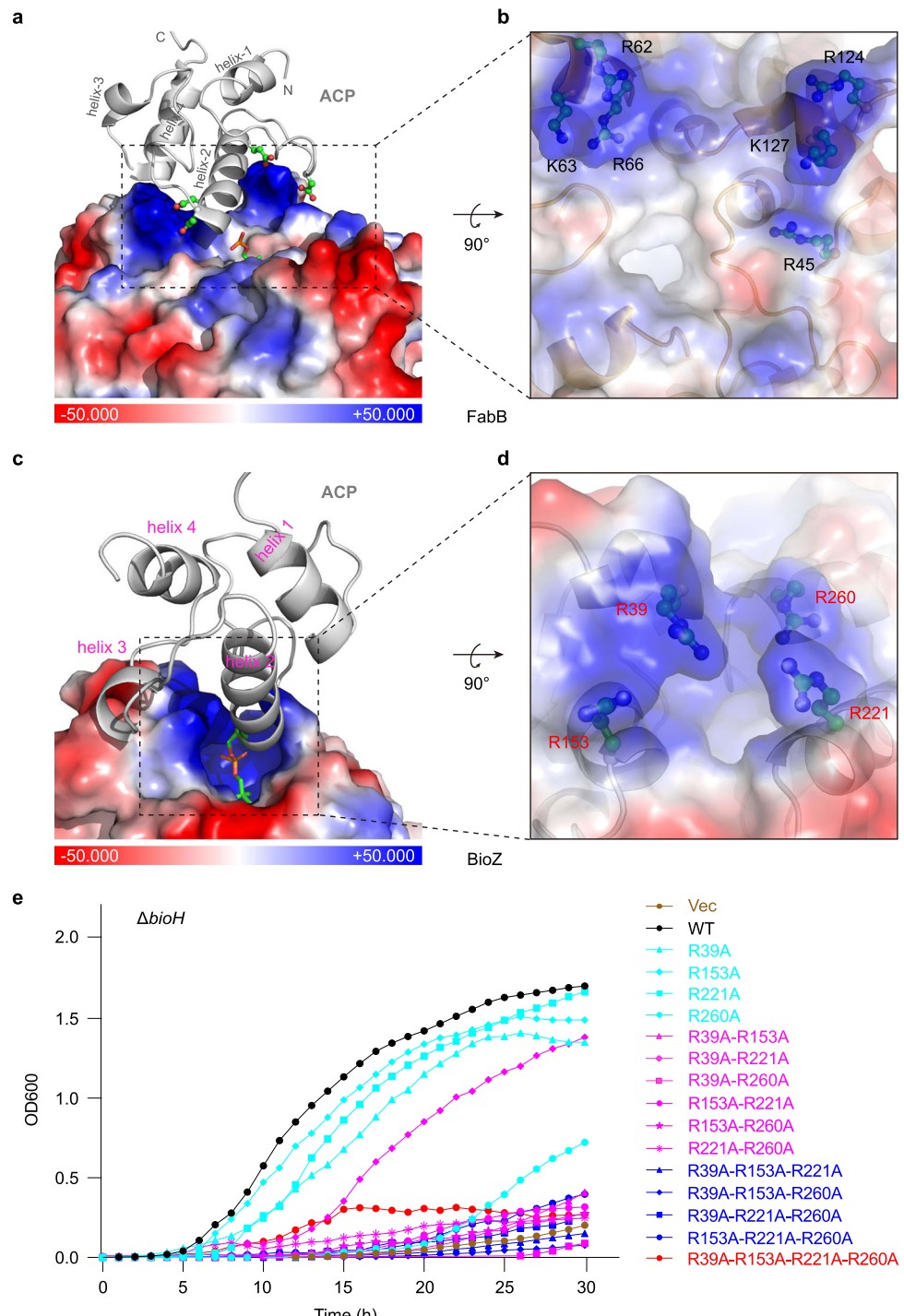

**Fig. 7 Structural insights into the recognition of malonyl-ACP by the BioZ enzyme. a** Structural display of the ACP moiety of acyl-ACP substrate bound to the paradigm enzyme of FAS I, the *E. coli* FabB. **b** An enlarged view of the ACP-interacting interface of FabB that is constituted by six positively-charged residues. The image is generated through a counter-clockwise 90° rotation of the rectangle-dashed lined region (**a**). **c** Binding model of glutaryl-ACP to the positively-charged surface of AtBioZ. ACP is given in ribbon structure colored gray, which comprises four α-helices (helix-1 to helix-4). The interface of FabB (**a**) [and/or BioZ (**c**)] interacting with ACP group is illustrated in the surface electrostatic structure. The blue denotes positive charge, whereas the red refers to negative charge. **d** Visualization of the putative ACP-binding domain of BioZ. Presumably, it contains four basic residues, namely R39, R153, R221, and R260. This image is given through the counter-clockwise 90° rotation of the square-dashed lined region (**c**). **e** Growth curve-based assay to probe the role of the putative four basic residues-comprising, ACP-binding interface in the BioZ function. The strains used here were listed in Supplementary Table 1, and the growth curve was plotted as described in Fig. 5e. A representative result was given.

positively-charged amino acids, namely Arg39, Arg153, Arg221, and Arg260 (Fig. 7c, d). Among them, R39 and R153 was also predicted to interact with the CoA cargo. It is plausible that these residues play certain roles in the interaction of BioZ with the negatively-charged residues on ACP protein.

To test this hypothesis, we generated a number of BioZ mutants (Supplementary Table 1) and examined them in vivo. These mutants of BioZ are divided into 4 groups: (i) four single mutants (R39A, R153A, R221A, and R260A), (ii) six sets of double mutants (R39A/R153A, R39A/R221A, R39A/R260A, R153A/R221A, R153A/R260A, and R221A/R260A), (iii) four triple mutants (R39A/R153A/R221A, R39A/R153A/R260A, R39A/R221A/R260A, and R153A/R221A/R260A), and (iv) a quadruple mutant (R39A/R153A/R221A/R260A). Next, genetic complementation of the ΔbioH biotin-auxotrophic strain was performed using pBAD24-borne bioZ mutants (Fig. 7e). As a result, (i) only one of four single bioZ mutants (i.e., R260A) lost full activity as illustrated in the phenotypic growth curves of ΔbioH; (ii) nearly all the six double mutants of bioZ exhibited poor activities; (iii) all the four triple mutants of bioZ were functionally impaired; and finally, the quadruple mutant of bioZ (R39A/R153A/R221A/R260A) was inactive (Fig. 7e). Together with our structural data, these in vivo results defined a functional interface between BioZ and malonyl-ACP, which is important in bypassing the canonical earlier steps of biotin synthesis.

**Recognition of primer glutaryl-CoA by BioZ.** The biochemical data indicated that BioZ exploits glutaryl-CoA and glutaryl-ACP both as the primer substrates in the in vitro reconstituted FAS reaction system (Fig. 3 and Supplementary Fig. 10). In contrast to those of malonyl-ACP (Fig. 6e) and holo-ACP alone (Supplementary Fig. 18), glutaryl-ACP exhibited more tight binding the AtBioZ(C115A) mutant enzyme, as determined by ITC experiments with the stoichiometry of $N = 0.978 \pm 0.028$, and $Kd = 6.167 \pm 0.068\ \mu M$ (Supplementary Fig. 11). It seemed likely that BioZ enzyme can adjust (perhaps expand) the configuration of primer-loading tunnel to accommodate primer substrate of longer chain acyl-ACP (CoA). To address this issue, we thus applied AutoDock Vina-based molecular docking. As expected, the ACP-removed glutaryl moiety (i.e., glutarate) we selected was well docked into the AtBioZ enzyme (Supplementary Fig. 16a-b). Structural superposition revealed that this pocket occupied by glutarate is conserved amongst BioZ and its ancestor FabH enzymes (Supplementary Fig. 16c). This indicated that the primer-loading tunnel on BioZ is an evolutionary relic. Very recently, Hu and Cronan reported genetic evidence for glutaryl-CoA originating from bacterial lysine catabolism, and argued the feasibility of glutaryl-ACP as a physiological primer of BioZ[47]. Because that both CoA and ACP are acidic molecules carrying a flexible Ppan moiety, it is not unusual that CoA and ACP mimic one another as noncognate substrates in the context of fatty acid metabolism. If so, this might explain why the above two C5-thioesters (C5-CoA and C5-ACP) can be recognized by BioZ in vitro and displayed comparable enzymatic affinity.

Since glutaryl-CoA is a most-likely intracellular primer/ligand, we then ask the question how the substrate-tunnel is tolerant with its free charged ω-carboxyl group. Subsequently, molecular docking of an intact glutaryl-CoA into AtBioZ was carried out (Fig. 8a). This allowed us to visualize a fine substrate-loading tunnel accessible to glutaryl-CoA (Fig. 8b). Not surprisingly, our ITC analyses demonstrated an efficient binding of glutaryl-CoA to BioZ in which the stoichiometry appears as follows: $N = 0.80 \pm 0.01$, and $Kd = 2.64 \pm 0.23\ \mu M$ (Fig. 8c). This was quite similar to the scenario seen with glutaryl-ACP (Supplementary Fig. 11). Indeed, it in turn verified the reliability in molecular docking of BioZ/glutaryl-CoA. Structural analysis of the BioZ tunnel surface

localized three residues of interest that might interact/neutralize the free carboxyl group of glutaryl-CoA substrate. They included Serine 84 (S84), Arginine 147 (R147), and Serine 287 (S287), respectively (Fig. 9a). Moreover, sequence alignments showed that all the three putative sites are extremely conserved across different BioZ homologs throughout α-proteobacteria (Fig. 9b). Thereafter, we applied alanine substitution to generate three single mutants (S84A, R147A, and S287A), and tested the in vivo roles in bypassing the need of BioH in demethylation of methyl-pimeloyl-ACP in E. coli. Although the R147A mutant of bioZ retained partial activity, the other two mutants (S84A and S287A) cannot allow the ΔbioH biotin-auxotrophic strain to well appear on the nonpermissive condition, biotin-deficient M9 minimal agar plates (Fig. 9c). A similar scenario was observed with growth curves of the aforementioned strains in the liquid M9 media lacking biotin (Fig. 9d). In addition, our structure-guided functional assays demonstrated that four additional residues (E152, T216, M217, and N258) play roles in the transient crosstalk/stabilization of BioZ with the CoA cargo. Taken together, these results constituted functional definition of the primer substrate-loading tunnel within the AtBioZ enzyme.

## Discussion

Along with recent findings by other two research groups[20–22,33,34], the data we reported here contributes important biochemical and structural information of de novo biotin synthesis, which proceeds using three distinct pathways: (i) BioC-BioH pathway[20–22], (ii) BioI-BioW pathway[33,34], or (iii) noncanonical BioZ pathway (Fig. 3). Among them, the BioC-BioH pathway (and/or its equivalent, like BioJ[27,42]) is the prevalent route. In contrast, the BioZ pathway seems to be restricted to the phylum of α-proteobacteria (Supplementary Figs. 1-4). The exclusive occupation of BioZ in certain α-proteobacterial species, like A. tumefaciens, can be in part explained by its origin and evolution of FabH in the close relatives of Rhizobiales (Fig. 2). Probably, a long-term selection and gradual evolution domesticated this FabH-like enzyme, BioZ to do an "extra and/or redundant job", providing a substrate for biotin synthesis (Fig. 3 and Supplementary Fig. 9). To less extents, the three residue-forming catalytic triad (C115, H255, and N285) of BioZ is configurationally similar to that of FabB (C163, H298, and H333, in Supplementary Fig. 19a-d). This hypothesis is proved by our discovery that BioZ ligates the primer glutaryl-CoA (ACP) with the extension unit malonyl-ACP to give 5-keto-pimeloyl-ACP (converted by a round of FAS II cycle into pimeloyl-ACP, Supplementary Fig. 9), a bona fide precursor for the latter steps of biotin synthesis.

Of note, during the revision of this manuscript, Hu and Cronan reported a genetic and enzymatic study of BioZ action, pointing out lysine catabolism as the source of glutaryl-CoA, an intracellular priming substrate for BioZ[47]. In generally consistent with the description of Hu and Cronan[47], we presented integrative evidence in vivo and in vitro (covering bacterial genetics, biochemistry and chemical biology) for BioZ activity (Fig. 3 and Supplementary Figs. 10-14). In addition to its origin (Fig. 2), we also provided structural basis for BioZ action in condensing glutaryl-CoA (ACP) with malonyl-ACP (Figs. 4, 5). More importantly, structure-to-function study of BioZ allowed us to finely define the primer glutaryl-CoA substrate-loading tunnel (Figs. 8, 9), as well as the electrostatic interface recognized by the extension unit malonyl-ACP (Figs. 6, 7). The accumulated data supported the proposal that BioZ utilizes a sequential substrate-binding mode for two substrates (one is the primer glutaryl-CoA (ACP) and the other denotes an extension unit, malonyl-ACP) (Supplementary Figs. 18 and 20). Similar to other β-ketoacyl-ACP-synthases, such as FabB[46], it is mechanistically reasonable to

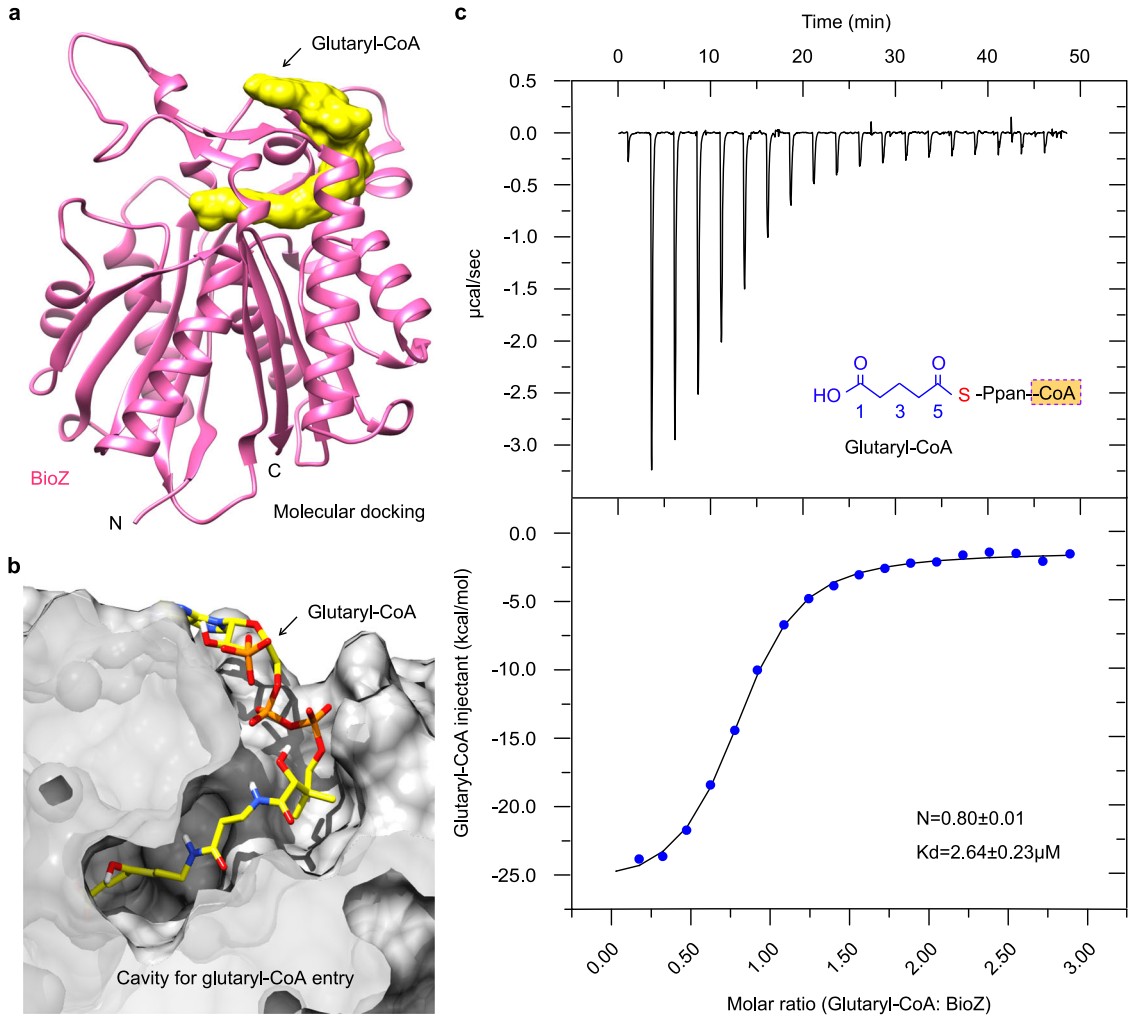

**Fig. 8 Biochemical evidence that BioZ binds to its substrate glutaryl-CoA. a** Molecular docking of BioZ with glutaryl-CoA. **b** Structural snapshot of the glutaryl-CoA substrate-loading tunnel within BioZ enzyme. **c** Use of ITC to measure the binding of glutaryl-CoA to BioZ protein. A representative ITC result from three independent experiments is displayed. Thus, the stoichiometry values (N and Kd) are expressed in an average ± SD. The stoichiometry of glutaryl-CoA binding to BioZ enzyme is proposed to be 1:1, in that the molar ratio was measured to be around 0.8. The parameter of binding affinity (Kd) is 2.64 ± 0.23 μM.

assume that dimeric BioZ enzyme successively binds to the aforementioned two substrates via surface electrostatic interaction at the same ACP-binding sites (Figs. 5, 6). Given that malonyl-ACP plays a critical role in the elongation steps catalyzed by β-ketoacyl-ACP-synthases, we therefore formulate a working model for BioZ action, in which BioZ likely binds to glutaryl-CoA (ACP), then malonyl-ACP (Supplementary Fig. 20). In this model, a glutaryl-loaded CoA (ACP) first associates with BioZ, the active site of which, cysteine (Cys115) attacks the glutaryl moiety and releases the CoA (ACP) carrier. Then, the glutaryl-BioZ intermediate is ligated with a malonyl-ACP, leading to the addition of the acyl chain of glutaryl moiety with two carbon units (from C5 to C7). Finally, the 5-keto-pimeloyl ACP dissociates from the dimeric BioZ enzyme, which is then modified by FAS II system, prior to entry into the latter steps of biotin synthesis (Supplementary Fig. 9). That is the reason why the *bioZ* gene bypasses the genetic requirement of "*bioC-bioH*" in *E. coli* on the nonpermissive growth condition (Figs. 5e, f and 9c, d). It is unusual, but not without any precedent, that the activity of BioZ originates from its noncognate substrate preference/specificity in the context of FAS II fatty acid synthesis. Therefore, BioZ essentially serves as a noncanonical player within the family of β-ketoacyl-ACP synthase III, the enzymatic action of which

satisfies the physiological demand of biotin in a number of *bioC*- and *bioW*-lacking bacteria (Supplementary Fig. 1). Structural comparison of different β-ketoacyl-ACP synthase III suggests that the gain of an atypical FabH-like activity of BioZ might be due to its distinct lid domain rather than the conserved catalytic domain (Fig. 3). This hints that an evolutionary advantage occurs in engineering a redundant (and/or duplicated) gene, but not creating an additional gene for a previously-unrecognized metabolic ability.

Because that biotin is recognized as a restriction (and/or nutritional) virulence factor[42,48–50], the action of BioZ is presumably to participate into successful infection of the plant pathogen *A. tumefaciens*[39] and the human pathogen *B. melitensis*[36]. Structure-guided functional dissection of both catalytic triad (Fig. 4) and substrate-loading tunnel (Fig. 6) might render BioZ as an anti-virulence drug target, which can be utilized into the screen of lead compounds and design of inhibitors of small molecules. Given that *A. tumefaciens*, the *bioZ*-bearing microbe, exhibits the potential of profligate synthesis of biotin (perhaps its precursor DTB) in the excess of its physiological demand[39], it is of much interest to ask the three questions bellowed: (i) if or not the excessive secretion of biotin/DTB is an ecologically selective outcome, benefiting those biotin-auxotrophic bacterial species inhabited in the same niche;

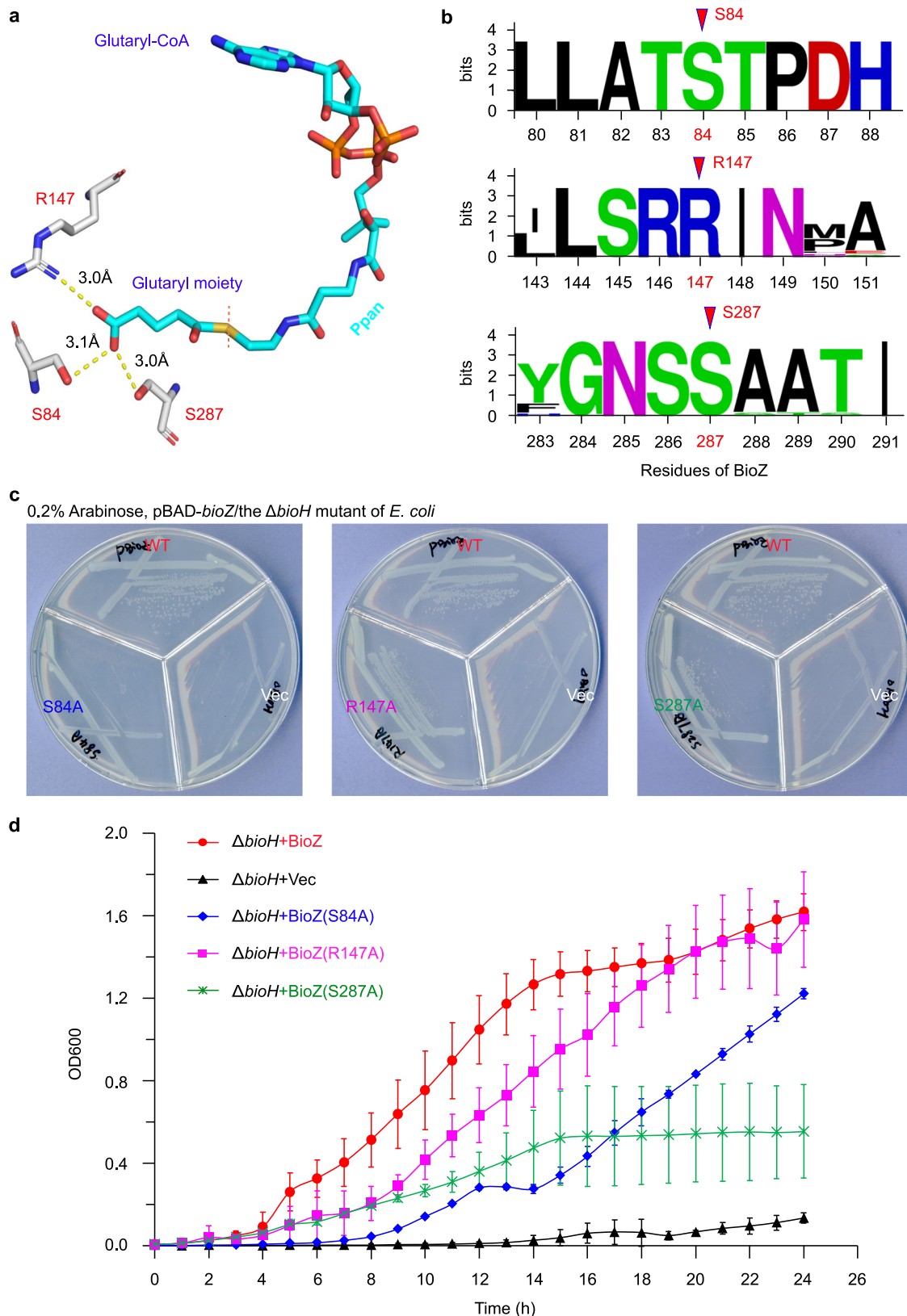

(ii) if *A. tumefaciens* can be developed into a 'green chemistry' tool for biotin production; and (iii) whether or not biotin production is greatly elevated in the BioZ-engineered *E. coli* strains? Along with the genetic & enzymatic study of Hu and Cronan[47], our report extends biochemical and structural insights into molecular mechanism for bacterial BioZ biotin synthesis. To the best of our knowledge, the BioZ pathway represents a third machinery route for the formation of the biotin precursor pimeloyl-ACP, and poses extensively promising implications in the agricultural, pharmaceutical, and biomedical fields.

**Fig. 9 Functional dissection of the glutaryl-CoA substrate-loading tunnel within the BioZ enzyme. a** The proposed critical residues of BioZ contributing to the neutralization of free carboxyl group of glutaryl-CoA. Three residues that might neutralize the charge of the free carboxyl group from glutaryl-CoA is determined by molecular docking to localize at the distant end of substrate-loading pocket (Fig. 8a, b). Namely, they are Serine 84 (S84), Arginine 147 (R147), and Serine 287 (S287), respectively (**a**). **b** The three possible neutralizing residues (S84, R147, and S287) are conserved across all the BioZ homologs from different α-proteobacteria. Sequence logo was generated using the server WebLogo (http://weblogo.berkeley.edu/logo.cgi). **c** Use of site-directed mutagenesis to assay the varying roles of the three putative charge-neutralizing residues in BioZ activity. **d** Growth curves of the ΔbioH strains carrying either the wild-type bioZ or its point-mutants. Derivatives of the ΔbioH biotin-auxotrophic strain (Supplementary Table 1) were assayed on the nonpermissive growth condition of M9 minimal media without any biotin. The addition of 0.2% arabinose induced the expression of pBAD24-borne bioZ (and/or its point-mutants).

## Methods

**Phylogenetic analysis of BioZ.** The genetic context of bioZ-like loci and genes coding for FabH, FabB, and FabF in bacterial genomes were comparatively investigated in NCBI. RefSeq archived 334 α-proteobacteria genomes by using BLASTp-based searches. BioZ homologues were predicted by using BLASTp and BLASTn, respectively, against protein and nucleotide sequences of biotin synthesis gene clusters (bioBDFAZ + bioMN) from Agrobacterium tumefaciens str. C58 and Brucella melitensis bv. 1 str. 16 M[51]. Then, the functional relatedness of neighboring genes was carefully examined and correctly placed in the context of lipid metabolism[37,42,52]. In light of respective prototypes of FabH[53], FabF[54], and FabB[54,55], the homology search identified 106 homologs of KAS enzymes, consisting of 50 FabH, 41 FabF, and 15 FabB. Finally, the phylogenetic relationships between BioZ and its homologs (FabH, FabB, and FabF) were calculated with the software of MEGA7[56].

**Genetic manipulations.** To address the function of the BioZ enzyme, three bioZ homologs of α-proteobacteria (A. tumefaciens, B. melitensis, and Rhizobium) were amplified by PCR and/or synthesized in vitro. The bioZ genes of these origins were cloned into an arabinose-inducible pBAD24 expression vector, giving recombinant plasmids such as pBAD24::atbioZ (Supplementary Table 1). To probe whether BioZ bypasses the cognate earlier steps of biotin synthesis in E. coli, the recombinant plasmids were transformed into a single mutant ΔbioC (or ΔbioH) or into the double mutant ΔbioC/ΔbioH (Supplementary Table 1). Similarly, all the three versions of bioZ were cloned into an isopropyl-β-D-thiogalactopyranoside (IPTG)-inducible expression pET28a vector, to produce a soluble form of BioZ proteins suitable for in vitro enzymatic assays and crystallization screens. Among them, only pET28::bioZ from A. tumefaciens (Supplementary Table 1), was found to produce soluble BioZ protein. Additionally, the fabH1 of A. tumefaciens was also cloned into pET28 vector to prepare its soluble protein. To generate distinct site-directed point mutations of bioZ, two different techniques were used accordingly: (i) primers-based overlapping PCR (Supplementary Table 2), and (ii) the Mut Express II fast mutagenesis kit V2 (Vazyme Biotech Co., Ltd.). All plasmids generated in this study were confirmed by PCR assays (Supplementary Table 2), followed by direct DNA sequencing. When necessary, antibiotics were supplemented as follows: 50 mg/ml for Ampicillin and 30 mg/ml for Kanamycin.

**Protein expression, purification, and identification.** The prokaryotic system of E. coli BL21 with pET28::atbioZ was used to produce the BioZ protein (Supplementary Table 1). Protein expression and purification (small batches) was first optimized with GF buffer (25 mM Tris-HCl (pH 8.0), 100 mM NaCl, and 2 mM DTT), prior to a large-scale production. As described for BioJ protein with little change[27,42], the bacterial lysate containing six histidine-tagged AtBioZ protein was prepared with GF buffer, and then subjected to the purification by affinity chromatography using a Ni-NTA column (Qiagen). TEV enzyme was used overnight to cleave the N-terminal hexa-histidine (6× his) tag, followed by size exclusion chromatography purification using Superdex 200 Increase 10/300 column (GE Healthcare). The purity of the collected samples was tested on SDS-PAGE. Following trypsin digestion, the identity of the purified protein was verified using Matrix-Assisted Laser Desorption/ionization Time of Flight (MALDI-TOF) mass spectrometry[51]. The purified AtBioZ protein was concentrated to 15 mg/ml with the GF buffer (25 mM Tris-HCl (pH 8.0), 100 mM NaCl and 2 mM DTT) for further crystallization screens. All the other protein components used in our in vitro system of fatty acid synthesis were routinely prepared as previously reported[57]. In addition to the E. coli holo-ACP[58] and the Vibrio harveyi fatty acid ACP synthetase AasS[59], six enzymes of E. coli were also used, namely malonyl-CoA: ACP trans-acylase (FabD)[60], β-ketoacyl-ACP reductase (FabG)[61], β-hydroxyacyl-ACP dehydratase/isomerase (FabA/FabZ)[54,55], enoyl-ACP reductase (FabI)[62], and β-ketoacyl-ACP synthase III (FabH)[53]. To compare the substrate specificity of AtBioZ (FabH2), we also introduced FabH1 of A. tumefaciens (AtFabH1), which was expressed and purified routinely as performed with BioJ and BioZ here.

**Isothermal titration calorimetry.** To determine the stoichiometry of BioZ to varieties of substrates, the experiments of isothermal titration calorimetry (ITC) were performed using the microcalorimeter (MicroCal PEAQ-ITC). The BioZ-binding partners/substrates examined here included glutaryl-ACP (prepared in this study), glutaryl-CoA (Sigma–Aldrich), malonyl-ACP, and ACP alone. In particular, AasS catalyzed the reaction of methyl-glutaryl-ACP synthesis from holo-ACP and monomethyl-glutaric acid (Sigma–Aldrich). Glutaryl-ACP was prepared with the methyl removal of methyl-glutaryl-ACP by the BioJ enzyme (Supplementary Fig. 7b)[27,42]. All the measurements were conducted at room temperature (25 °C). Similar to the preparation of the complex of BioH and methyl-pimeloyl-ACP[22], the mutant version (C115A) of BioZ that lacks enzymatic activity, but retains substrate-binding ability, were prepared in the buffer containing 20 mM Tris (pH 8.0) and 150 mM NaCl. Then, the purified BioZ(C115A) protein (50 μM) was loaded into the reaction wells. In every binding assay, the aforementioned four substrates were consistently at the concentration of 750 μM in the same buffer, and titrated with a syringe into the BioZ (C115A) protein samples. In general, the titration was consisted of 19 injections of 2.0 μl each 150 s (except for the first injection of 0.4 μl). Three independent measurements of ITC were included. The binding affinities were calculated by fitting the integrated titration data with a one binding-site model in Origin 7.0 software. The values of N and Kd (dissociation parameter) were expressed in an average ± standard deviation (SD).

**DTB/biotin bioassay.** Because that BioZ restores growth of ΔbioH on biotin-deficient media, a phenotype similar to that of BioJ[27,42], biotin bioassay was utilized to test the hypothesis that BioZ acts as a nonhomologous isoenzyme of BioJ. Biotin bioassay was essentially carried out as we recently described with BioJ[27]. The system of biotin (or DTB, a biotin precursor) was reconstituted in vitro as described earlier by Lin et al.[20]. Briefly, to trigger DTB synthesis, the purified protein AtBioZ (or the positive control BioJ) was supplemented into the cell-free crude extract of STL24 (E. coli ΔbioH)[25]. Then, the production of DTB or biotin was examined using a biotin indicator strain ER90 (ΔbioF ΔbioC ΔbioD)[20]. The release of an insoluble red pigment caused by the reduction of 2,3,5-triphenyl tetrazolium chloride (TTC) indicate the supply of biotin (and/or DTB) produced in the in vitro system[42]. In addition, DTB/botin assays were also applied to confirm the production and secretion of biotin/DTB of the wild-type of A. tumefaciens NTL4, as well as the need of biotin for the A. tumefaciens biotin-auxotrophic strain (ΔbioBFDA)[39].

**Assay of BioZ activity in vitro.** To test BioZ activity, the system of fatty acid synthesis in vitro was reconstituted as described earlier by Zhu and Cronan[57]. Holo-ACP of E. coli was prepared routinely[58,63]. Prior to the reaction of fatty acid synthesis, certain substrates of fatty acyl-ACP thioesters were synthesized in vitro. Unlike that malonyl-ACP (Mal-ACP, C3-ACP) was synthesized by FabD from 5 mM malonyl-CoA (Sigma–Aldrich) with holo-ACP, hexanoyl-ACP (Hex-ACP, C6-ACP) was synthesized using the AasS-based ligation of holo-ACP with 5 mM hexanoyl-CoA (Sigma–Aldrich). In fact, AasS can also attach the two unusual substrates [methyl-glutaric acids (M-Glu) and methyl-pimelate (M-Pim)] to holo-ACP, giving M-Glu-ACP and M-Pim-ACP, respectively (Supplementary Fig. 7b)[59]. Subsequently, glutaryl-ACP (Glu-ACP, C5-ACP) and pimeloyl-ACP (Pim-ACP, C7-ACP) were separately produced via the BioJ-based cleavage of methyl moiety from M-Glu-ACP and M-Pim-ACP, respectively[27,42]. All the other acyl-CoA thioesters tested here are products purchased from Sigma–Aldrich. Namely, they included acetyl-CoA (C2-CoA), butanoyl-CoA (C4-CoA), glutaryl-CoA (Glu-CoA, C5-CoA), hexanoyl-ACP (Hex-CoA, C6-CoA), and octanoyl-CoA (Oct-CoA, C8-CoA). In brief, the reaction system (50 μl in total) contains 0.1 M Tris-HCl (pH 8.0), 5 mM DTT, 200 μM NADH, 200 μM NADPH, 100 μM malonyl-CoA, 0.2 μg/μl holo-ACP, 100 μM fatty acyl-CoA, 0.1 μg of E. coli FabD, FabG, FabA, and FabI enzymes. Upon the addition of 0.1 μg of E. coli FabH (control) or AtBioZ into the aforementioned reaction system, it was incubated at 37 °C for 1 h. Then, the resultant acyl-ACP products were detected with conformationally sensitive, poly-acrylamide gel electrophoresis (PAGE) (17.5%, pH9.5) containing 0.5 M urea. Of note, pimeloyl-ACP acted as the positive control that was generated through the removal of methyl group by BioJ from methyl-pimeloyl-ACP.

**LC-MS/MS identification of fatty acylation of ACP.** To identify the fatty acyl modification of ACP, liquid chromatography (LC) mass spectrometry was used as described earlier for the acetylation of BioQ with minor changes[52]. The reaction mixture of the in vitro reconstituted system for AtBioZ-involved fatty acid synthesis were separated with the electrophoresis of conformationally sensitive, 0.5 M urea PAGE (17.5%, pH9.5), and then the protein band of interest was cut from the gel, and digested with pepsin (rather than the routine trypsin). In

particular, the positive control pimeloyl-ACP was verified with MALDI-TOF. The mixture of resultant peptides was injected into the trap column with a flow rate of 10 μl/min for 2 min using a Thermo Scientific Easy nanoLC 1000. The chromatographic system was composed of a trapping column (75 μm × 2 cm, nanoviper, C18, 3 μM, 100 Å) and an analytical column (50 μm × 15 cm, nanoviper, C18, 2 μM,100 Å). Data collection was performed using Thermo LTQ-Orbitrap Velos Pro equipped Nanospray Flex ionization source and FTMS (Fourier transform ion cyclotron resonance mass analyzer) analyzer combined with Thermo LTQ-Orbitrap Elite equipped Ion Trap analyzer. The parameters for FTMS were as follows: Data collection were at 60 K for the full scan MS, positive as polarity, profile as data type, and then proceeded to isolate the top 20 ions for MS/MS by CID (1.0 $m/z$ isolation width, 35% collision energy, 0.25 activation Q, 10 ms activation time). Scan Range was set as 300 $m/z$ First Mass and 2000 $m/z$ Last Mass. The parameters for Ion Trap analyzer were normal mass range, rapid scan rate, and centroid data type. Among the MS spectrums of peptides detected by Thermo Fisher Orbitrap elite mass spectrometer (Thermo Scientific, USA), a number of peaks were attributed to the C7-fatty acylation of ACP, following the database search with the of software Proteome Discover 2.0[64].

**Crystallization, data collection and structural determination**. In brief, 0.6 μl of AtBioZ protein (~15 mg/ml) mixed with 0.6 μl of the screening buffer was subjected to the routine screen of crystallization using the hanging drop vapor diffusion method. Five different screening kits were used: (i) Crystal screen I & II (Hampton Research), (ii) Index comprising 96 reagents (Hampton Research); (iii) PEG/ion, a kit of high purity Polyethylene glycol 3350 in combinations with 48 unique salts (Hampton Research); (iv) PEGRx, a polymer- and pH-based crystallization screen (Hampton Research); and (v) WIZAED screen (Rigaku). As a result, crystals with good diffraction were grown in 0.2 M sodium nitrate and 20% (v/v) PEG 3350 at 16 °C on the 3rd day post-screen. The crystals were harvested and flash frozen in liquid nitrogen with 20% glycerol as a cryoprotectant. Collection of X-ray diffraction data were performed at BL17U1 beamline of Shanghai Synchrotron Radiation Facility (SSRF). Diffraction images were calculated by HKL-2000 program[65]. The initial model was solved by molecular replacement (MR) using the structure of Staphylococcus aureus FabH (PDB: 1ZOW) as the searching model[66,67]. Model building and crystallographic refinement were conducted with COOT and PHENIX[68,69]. The final structure of AtBioZ was solved at 1.99 Å (Table 1) and deposited into the Protein Data Bank (PDB) with the accession entry: 6KUE.

**Molecular docking between BioZ and substrates**. The protein three-dimensional structures were obtained from RCSB PDB database[70]. Structure superposition was performed using MatchMaker tool of UCSF Chimera software[71]. The structure images were generated using UCSF Chimera and PyMOL software (https://pymol.org). BioZ possesses similar fold and active site structure as FabB. We built a schematic model of BioZ-ACP complex based on ClusPro server-aided molecular docking[72]. Given that (i) three negatively-charged amino acids (Asp35, Asp38, and Glu47) have roles in the interaction between FabB and ACP, and (ii) FabB accesses its substrate by interacting with ACP protein[46], four arginines (Arg39, Arg153, Arg221, and Arg260) of BioZ were located in the proximity of its substrate-loading entrance.

To further dissect binding model of the primer glutaryl-CoA (or glutarate alone) and BioZ, molecular docking was applied with AutoDock Vina (version 1.1.2)[73]. Apart from the high-resolution (1.992 Å) 3D structure of BioZ (PDB ID: 6KUE), the glutaryl-CoA structure was extracted from a of glutaryl-CoA dehydrogenase/glutaryl-CoA complex structure (PDB ID: 3MPI, glutaryl-CoA identifier: GRA)[74]. All the water molecules in BioZ structure were removed. Polar hydrogens were added and Gasteiger atomic partial charges were assigned to both BioZ and glutaryl-CoA using the AutoDockTools (version 1.5.7)[75]. All the rotatable bonds in ligand glutaryl-CoA were set to be flexible and all the protein residues of BioZ were set to be rigid during the docking procedure. The search space for docking was set to a 36 × 42 × 26 Å (center_x = −8.556, center_y = 30.342, center_z = 47.567) cube box, which surrounds the substrate-binding pocket and tunnel of the BioZ monomer. The binding model with best score from the docking results was selected for subsequent analysis. The images for interpreting the binding and residue interactions of BioZ and Glutaryl-CoA were generated by Pymol (version 2.5.0) (https://pymol.org/2/) and UCSF Chimera (version 1.14)[71].

**Reporting summary**. Further information on research design is available in the Nature Research Reporting Summary linked to this article.

## Data availability
All relevant data are available from authors. Besides the data in tables and figures, supporting information is accompanied with this paper. The accession number for the atomic coordinates of BioZ protein reported in this paper is PDB: 6KUE. Source data are provided with this paper.

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

## Acknowledgements

This work was supported by National Key R&D Program of China (2017YFD0500202, Y.F.), National Natural Science Foundation of China (31830001, 31570027, and 81772142, Y.F. and 31770948, S.O.), the Fundamental Research Funds for the Central Universities (3102020smxy002, T.C.), the Special Open Fund of Key Laboratory of Experimental Marine Biology, Chinese Academy of Sciences (SKF2020NO1, S.O.), Marine Economic Development Special Fund of Fujian Province (FJHJF-L-2020-2, S.O.), the Fujian Provincial Department of Science and Technology (2020Y4007 and 2021H0004, S.O.), and the High-Level Personnel Introduction Grant of Fujian Normal University (Z0210509, S.O.). Y.F. is a recipient of the national "Young 1000 Talents" award of China. We would like to thank Dr. Youping Xu (Analysis Center for Agro-biology and Environmental Sciences, Zhejiang University) for technical assistance in mass spectrometry (LC-MS/MS and MALDI-TOF MS), and the staffs at the beamline BL17U1 of Shanghai Synchrotron Radiation Facility (SSRF) for crystal data collection.

## Author contributions

Y.F. and S.O. designed and supervised this project; Y.F., S.Z., Y.X., H.G., T.C., Y.L., W.W., J.L., B.H., H.Z., and X.J. performed experiments; Y.F., Y.X., T.C., J.L., S.Z., W.W., and B.H. analyzed the data and prepared figures; H.Z., Y.X., X.J., and T.C. contributed to tools; Y.F., T.C., J.L., H.Z., H.G., S.O., Y.X., and B.H. drafted this manuscript.

## Competing interests

The authors declare no competing interests.
