## [Peer Review File · Nature Communications]

Reviewers' comments:

Reviewer #1 (Remarks to the Author):

Biotin is an important micronutrient whose early biosynthetic steps are very diverse. Thus far, three different pathways have been identified for synthesis of the pimelic acid precursor, namely the BioC/BioH pathway, the BioI/BioW pathway and the BioZ pathway. BioZ has been shown to complement *E. coli* BioH, analyzed to be a beta-ketoacyl-ACP synthase III (KAS III), and predicted to synthesize pimeloyl-ACP (Ref. 30), but its catalytic function has not been characterized. In the current work, the authors first successfully repeated the previous BioZ complementation of BioH (and also BioC) in *E. coli* and did a thorough analysis of the evolutionary origin of the protein. Subsequently, AtBioH was successfully expressed and biochemical assays were carried to demonstrate that the enzyme is indeed able to condense malonyl-ACP and glutaryl-ACP to form pimeloyl-ACP, thus convincingly establishing the catalytic function of the enzyme. Next, the enzyme was crystallized and its structure was solved at 1.99 Angstrom to reveal that it is indeed a KAS III resembling the many previously solved structures of other members in the same category. Finally, site-directed mutagenesis and functional assays were used to identify the catalytic triad and the structural motif (composed of four positive arginine residues) responsible for recognizing the ACP substrate. These results provide unambiguous evidence for the proposed biological function of BioZ and establish the structural basis for this function, although neither the catalytic function nor the molecular mechanism is new.

A caveat in the structural part of the work is the lack of a binding pocket responsible for recognition of the glutaryl moiety of the glutaryl-ACP substrate. This binding pocket is where BioZ differs from all other members of KAS and is thus suggested to be identified to allow a better understanding of the catalytic mechanism. This pocket could be readily recognized or speculated in the structural comparisons that had been carried in the work. The roles of the suspected amino acid residues could then be readily established by site-directed mutagenesis and the assays that were already used in characterization of the wildtype enzyme or the mutants.

The presentation of title and abstract gives the impression that the work discovers a new, alternative pathway for the biosynthesis of pimelic acid or biotin. For example, the abstract states that 'Here we report a new mechanism that BioZ bypass the canonical route to begin biotin synthesis.' This, however, is not true. As mentioned above, the BioZ pathway was essentially established by the previous study presented in Ref. 30. What the authors have done is to define the substrates and establish the catalytic function of the enzyme and to solve its crystal structure. It is also noted that the catalytic mechanism is also not new. Thus, both the title and abstract should be modified to more accurately report the findings. Writing of the other parts of the work is clear and easy to understand.

Reviewer #2 (Remarks to the Author):

This is an interesting and important study that puts yet another new and exciting spin on biotin biosynthesis. Namely, a "third way" of synthesis of the key C7 dicarboxylic acid pimelate precursor. Known pathways of C7 biosynthesis are the BioC/BioH pathway in *E. coli* and BioI/bioW in *Bacillus subtilis*.

This new BioZ pathway looks to be present in α -proteobacteria such as *Agrobacterium tumefaciens* (At), *Rhizobium* (Rh) and *Brucella melitensis* (Bm).

They present strong evidence to support the assignment of BioZ as a novel FabH-like ketosynthase (KS) enzyme that catalyses the decarboxylative condensation between C3 malonyl- thioester and C5 glutaryl-thioester (they suggest as acyl carrier proteins (ACPs) thioesters) to give the pimeloyl-thioester. They show that this occurs on an ACP – they used the *E. coli* ACP for these studies. They show that of the 3 BioZs they tried (Bm, Rhiz and At BioZ) both the Bm and Rhiz complement *E. coli* BioC and BioC/BioH mutant. They also determine the crystal structure of At BioZ to 1.99 Å res PDB code: 6KUE – to be released). It is a very interesting structure and compares well with the KS superfamily which has a conserved catalytic triad of Cys115, His255 and Asn285. They then go on to make mutants of these residues to confirm their roles in function and activity (C115A, H255A and N285A) in complementation, assays and growth curves. Figure 3 is very nice but I would render the catalytic triad residues in each of the four structures.

They also investigate the binding of ACP to BioZ by making models using other KS and ACP complexes. They suggest that BioZ uses C3 malonyl- ACP and C5 glutaryl-ACP as substrates. Does this suggest that binds two ACPs at once? They use the recent FabB-ACP complex (PDB: 5KOF) from Burkart and Tsai (Nature Chem Biol, 2019) to model how the BioZ:ACP complex forms. They should make sure they state that this interesting and very useful complex is a chemically-crosslinked complex between *E. coli* Fab and ACP.

Fig. 5 needs to be re-drawn so that the two complexes have the same relative orientation – I would suggest they overlay on the two catalytic triads – please fix/centre on the conserved cysteines (FabH has C163, H333, H298).

They use this model to identify possible Arg residues that are involved in ACP binding – mutational analysis of these 4 Arg residues supports this.

Throughout they use Ec ACP but there is no discussion about what % conservation there between the Ec ACP and *Agrobacterium tumefaciens* ACP? This should be discussed.

Overall – an interesting paper and important addition – the paper needs to be proof read and English corrected by a native English speaker.

Other points.

Discussion: The biotin part should be in the introduction. The discussion should describe the mechanism – does glutaryl-ACP bind first and transfer glutarate to the Cys residue? Can they capture this? Then it should bind malonyl-ACP, decarboxylate then catalyse the condensation between the C5 and the C2 to give pimeloyl-ACP. Please expand this.

Fig.3 – nice 4 structures - how were they orientated? The label is wrong. EcBioF? Please label N- and C-termini.

3 BioZ homologs – show sequence alignment of these 3.

Is there an expt to show that BioZ does not catalyse the reaction between glutaryl-CoA and malonyl-CoA to give pimeloyl-CoA? It is referred to but no data.

General comments

FabH/ACP and FabZ/ACP alignment – orientate wrt catalytic triad?

Could not find Table S1 – please make sure it is available.

Of 3 only AtBioZ is soluble. His tag on N- term then removed.

Details of Ec ACP, FabG, FabA and FabI, AasS, BioH?? Gel of purified proteins – Fig S7 – missing BioH?

Fig S7b – no pimeloyl-ACP on gel as marker.

Fig. S8 is mass spectrometry and MS/MS analysis of ACP. This needs a Table of predicted and experimental values!!!! It is very difficult to interpret MS/MS data without a table.

Fig. S10a is the crystal structure of the At BioZ dimer – it has electron density for glutarate and glutarate is labelled in green – was glutarate bound? There is no discussion about this – please expand!

Reviewer #3 (Remarks to the Author):

In this manuscript, the authors claim to have elucidated a third route to access the pimeloyl-ACP, an intermediate en route the biosynthesis of biotin. Biotin biosynthesis is fragmented across bacteria, this is abundantly clear, and this manuscript attempts to answer a question that will be of broad interest to the community. This reviewer is not competent to judge the phylogenetic work and will restrict his comments to the enzymology and structural aspects of this work, in addition to the writing, all three of which are severely lacking. The following are the major deficiencies in this work:

1. Major language editing is required. Several sentences, particularly in the results section make no sense, at all. The figure legends need to be descriptive. The reviewer is completely unable to understand what is being shown in Figure S8, much of the biochemical validation rests on that figure and it is impossible to understand what is implied there. If the authors haven't done so already, they are encouraged to recruit a professional manuscript preparation service. Do not insert figure callings repeatedly in the middle of sentences. Abbreviations are not to be defined in figure legends, pymol does not need to be credited in figure legends, too many other instances to parse out.

2. Biochemical validation of BioZ activity: the assertion that BioZ accepts two ACP bound substrates needs to be validated using careful in vitro biochemical work. The biochemical validation, at the moment, is extremely shallow. At a minimum, the authors need to demonstrate whether or not BioZ possesses two distinct ACP

binding sites or not, what is the order of substrate binding, and if the substrate binding is sequential. MS1 based assays, without MS2 fragmentation, are not conclusive. In such a rich spectra containing innumerable ions, it is easy to "find what you are looking for" without accounting for alternate hypotheses. Given the data presented in this study, this reviewer is not convinced that the biochemical role of BioZ is as claimed. Contemporary analytical procedures do not justify the use of data shown in Fig. 2a, without securing firm standards of all species that they claim to detect, as conclusive.

3. BioZ structure description in the second half of Page 7 is entirely superfluous and can be omitted. The description of the catalytic triad can be simplified, a lot, Docking the ACP into the BioZ structure cannot be achieved using manual means only, crystal structures of both interaction partners are in hand, better computational rigor needs to be applied. Mutations of the basic residues can hit a lot of off-target things, again the authors are blind to alternate hypotheses. At the very least, the use of biophysical techniques such as ITC and/or SPR is essential to map out the ACP:BioZ binding stoichiometries. Manual modeling followed by in vivo testing does not justify the hypothesis here. Again, the reviewer does not find justification that two ACPs can bind to BioZ, or, that ACP binding can be sequential given the structural data presented. This is the crux of the biochemical activity, one ACP binds, transfers the payload to the active site, and then the second ACP binds. The structural data needs to support this hypothesis. At the moment, it does not.

Response letter

Reviewer #1

Overall comment: Biotin is an important micronutrient whose early biosynthetic steps are very diverse. Thus far, three different pathways have been identified for synthesis of the pimelic acid precursor, namely the BioC/BioH pathway, the BioI/BioW pathway and the BioZ pathway. BioZ has been shown to complement *E. coli* BioH, analyzed to be a beta-ketoacyl-ACP synthase III (KAS III), and predicted to synthesize pimeloyl-ACP (Ref. 30), but its catalytic function has not been characterized. In the current work, the authors first successfully repeated the previous BioZ complementation of BioH (and also BioC) in *E. coli* and did a thorough analysis of the evolutionary origin of the protein. Subsequently, AtBioH was successfully expressed and biochemical assays were carried to demonstrate that the enzyme is indeed able to condense malonyl-ACP and glutaryl-ACP to form pimeloyl-ACP, thus convincingly establishing the catalytic function of the enzyme. Next, the enzyme was crystallized and its structure was solved at 1.99 Angstrom to reveal that it is indeed a KAS III resembling the many previously solved structures of other members in the same category. Finally, site-directed mutagenesis and functional assays were used to identify the catalytic triad and the structural motif (composed of four positive arginine residues) responsible for recognizing the ACP substrate. These results provide unambiguous evidence for the proposed biological function of BioZ and establish the structural basis for this function, although neither the catalytic function nor the molecular mechanism is new.

Reply: We appreciate referee 1 for the overall summary and positive evaluation on the significant importance of this work in the context of biotin metabolism.

Q1: A caveat in the structural part of the work is the lack of a binding pocket responsible for recognition of the glutaryl moiety of the glutaryl-ACP substrate. This binding pocket is where BioZ differs from all other members of KAS and is thus suggested to be identified to allow a better understanding of the catalytic mechanism. This pocket could be readily recognized or speculated in the

structural comparisons that had been carried in the work. The roles of the suspected amino acid residues could then be readily established by site-directed mutagenesis and the assays that were already used in characterization of the wild-type enzyme or the mutants.

Reply: We do agree with this comment raised by Reviewer 1. We have combined molecular docking and site-directed mutagenesis to define this primer loading-tunnel (new_fig.8a-b). The binding of BioZ to glutaryl-CoA was demonstrated by our ITC experiment (new_fig.8c). Similarly, BioZ also binds to glutaryl-ACP efficiently in our ITC assays (new_fig.S11). More interesting, we localized that three crucial residues (S84, R147, and S287) at the bottom of this tunnel can neutralize the charge of free C-carboxyl group of glutaryl-CoA (ACP) (new_fig.9a-b), and functionally verified their roles in BioZ action in bypassing 'BioC-BioH' pathway in *E. coli* (new_fig.9c-d).

new_fig.8 Biochemical analysis for BioZ binding to the primer glutaryl-CoA

new_fig. 9 Functional identification of a glutaryl-CoA primer-loading tunnel of BioZ enzyme

Q2: The presentation of title and abstract gives the impression that the work discovers a new, alternative pathway for the biosynthesis of pimelic acid or biotin. For example, the abstract states that ‘Here we report a new mechanism that BioZ bypass the canonical route to begin biotin synthesis.’ This, however, is not true. As mentioned above, the BioZ pathway was essentially established by the previous study presented in Ref. 30. What the authors have done is to define the substrates and establish the catalytic function of the enzyme and to solve its crystal structure. It is also noted that the catalytic mechanism is also not new. Thus, both the title and abstract should be modified to more accurately report the findings. Writing of the other parts of the work is clear and easy to understand.

Reply: As the referee 1 indicated, Sullivan and coworkers found that a *fabH*-like gene, designated as *bioZ*, exists in the symbiosis island of *Mesorhizobium*, and restores the growth of the *E. coli* Δ *bioC* mutant in biotin-free conditions ¹. In fact, all the discovery is restricted to phenotypic relevance to biotin synthesis. However, our study presented biochemical & structural evidence for the definition of 3rd mechanism by which biotin synthesis begins. Of note, BioZ catalyzes the ligation of glutaryl-CoA (ACP) with malonyl-ACP to give 5-keto-pimeloyl-ACP. This intermediate can be converted by a round of FAS II route (FabG/A/I) into pimeloyl-ACP, a biotin precursor (new_Fig.S9). Additionally, we have rephrased accordingly. As for the title, it has been corrected into “Biochemical and structural characterization of the BioZ enzyme engaged in bacterial biotin synthesis pathway”. Also, the section of Abstract has been rephrased appropriately, removing the phrase “new” (on P1 of original version with changes tracked).

new_Fig.S11 A working model for the 3rd biotin synthesis pathway initiated by the BioZ enzyme

Reviewer #2

Overall comment: This is an interesting and important study that puts yet another new and exciting spin on biotin biosynthesis. Namely, a “third way” of synthesis of the key C7 dicarboxylic acid pimelate precursor. Known pathways of C7 biosynthesis are the BioC/BioH pathway in *E. coli* and Biol/bioW in *Bacillus subtilis*. This new BioZ pathway looks to be present in α -proteobacteria such as *Agrobacterium tumefaciens* (At), Rhizobium (Rh) and *Brucella melitensis* (Bm). They present strong evidence to support the assignment of BioZ as a novel FabH-like ketosynthase (KS) enzyme that catalyses the decarboxylative condensation between C3 malonyl- thioester and C5 glutaryl-thioester (they suggest as acyl carrier proteins (ACPs) thioesters) to give the pimeloyl-thioester. They show that this occurs on an ACP – they used the *E. coli* ACP for these studies. They show that of the 3 BioZs they tried (Bm, Rhiz and AtBioZ) both the Bm and Rhiz complement *E. coli* BioC and BioC/BioH mutant. They also determine the crystal structure of AtBioZ to 1.99Å res PDB code: 6KUE – to be released). It is a very interesting structure and compares well with the KS super-family which has a conserved catalytic triad of Cys115, His255 and Asn285. They then go on to make mutants of these residues to confirm their roles in function and activity (C115A, H255A and N285A) in complementation, assays and growth curves.

Reply: We do appreciate the reviewer 2 for the positive recognition of the importance of this study regarding the identification and functional definition of a 3rd pathway of pimelate precursor for biotin synthesis. Obviously, referee 2 gives a comprehensive & meaningful summary on the major conclusions.

Specific comments

Q1: Figure 3 is very nice but I would render the catalytic triad residues in each of the four structures.

Reply: According to the referee 2's suggestion, we have introduced the catalytic triad into each of the four structures (EcFabB, EcFabF, EcFabH and AtBioZ). Of note, the former Fig. 3 was renumbered with **Fig.4**, in which the catalytic triad separately denotes to C163, H298 & H333 in EcFabB; C164,

H304 & H341 in EcFabF; C112, H244 & N274 in EcFabH; and C115, H255 & N285 in AtBioZ (**new_Fig.4**).

new_Fig.4 Structural comparison of four FAS-type enzymes

Q2: They also investigate the binding of ACP to BioZ by making models using other KS and ACP complexes. They suggest that BioZ uses C3 malonyl- ACP and C5 glutaryl-ACP as substrates. Does this suggest that binds two ACPs at once? They use the recent FabB-ACP complex (PDB: 5KOF) from Burkart and Tsai (Naturer Chem Biol, 2019) to model how the BioZ: ACP complex forms. They should make sure they state that this interesting and very useful complex is a chemically-cross-linked complex between *E. coli* Fab and ACP.

Reply: It is a good comment. As reviewer 2 mentioned, we do use the FabB-ACP cross-linked complex (PDB: 5KOF) reported by Milligan *et al.*² to model the possible binding of BioZ to ACP (Fig. 6). Also, we have followed referee 2 suggestion to rephrase the statement into “chemically cross-linked complex” accordingly (on P10 of original version with changes tracked). The recent structures of four KAS enzymes cross-linked with ACP (FabF-ACP, FabB-ACP, FabZ-ACP & FabA-ACP) from Burkart’s research group^{2,3} visualized that dimeric KAS enzyme can be bound by two ACP (one on each monomer). BioZ is evolutionarily-relevant member within the KAS family enzyme, whose structure is determined to feature with a FabH-like dimeric configuration. Thus, we predict that BioZ might follow a similar rule in binding two ACP, resembling other paradigm KAS member, such as FabB. As referee 2 recommended, we have applied ITC to assay the stoichiometry of BioZ binding to three ACP species (C3-ACP, C5-ACP, and ACP alone, in Fig. 6e, S11 and S19). These results consistently verified that BioZ binds to ACP is at the ratio of 1:1. One of ITC results is given as follows:

new_Fig.6e ITC analysis of BioZ binding to C3-ACP

Q3: Fig. 5 needs to be re-drawn so that the two complexes have the same relative orientation – I would suggest they overlay on the two catalytic triads – please fix/centre on the conserved cysteine (FabB has C163, H333, H298). They use this model to identify possible Arg residues that are involved

in ACP binding – mutational analysis of these 4 Arg residues supports this.

Reply: In agreement with Reviewer 2, we also believe that it might be important to have the same relative orientation when comparing the two structures (FabB/ACP & BioZ/ACP) with certain similarity. Thus, we tried to generate such images for the above two complexes. While, we encountered some problems. In current situation, the catalytic triads are buried in the inner of the binding pocket of BioZ. In catalytic triads-centering orientation, the structure of ACP covered the most part of other three subunits (another ACP and BioZ dimer). A similar scenario is seen with the cross-linked complex of FabB and ACP. To avoid the difficulty we encountered, we alternatively attempted to create an additional figure (new_Fig.S20), which displays the catalytic triads centered in BioZ and FabB monomeric protein. Obviously, this figure is helpful for showing the similarity and difference of catalytic triads between BioZ and FabB.

new_Fig.S20 Structural insights into substrate recognition and catalysis of BioZ and/or FabB

a-b. Structural analysis of FabB crosslinked with ACP underscores three critical residues (C163, H298 & H333) implicated into the substrate recognition and catalysis

c-d. Structural snapshot of BioZ docked with ACP suggests three potential

residues (C115, H255 & N285) involved in the substrate recognition and catalysis

Q4: Throughout they use EcACP but there is no discussion about what % conservation there between the EcACP and *Agrobacterium tumefaciens* ACP? This should be discussed.

Reply: It is a useful point. We have added this information into the new fig. S6a. The similarity between the two ACP homologues from *E. coli* and *A. tumefaciens* is 71.8% (new_Fig.S6a).

new_Fig.S6a Sequence alignment of ACP protein of *E. coli* and *A. tumefaciens*

Q5: Overall – an interesting paper and important addition – the paper needs to be proof read and English corrected by a native English speaker.

Reply: Apart from our efforts to reorganize and improve the main text, an English native speaker, Dr. Hassan (a young biochemist in Stony Brook University) is involved in proof-reading as well as data analysis.

Other minor points

Q1: Discussion: The biotin part should be in the introduction. The discussion should describe the mechanism – does glutaryl-ACP bind first and transfer glutarate to the Cys residue? Can they capture this? Then it should bind malonyl-ACP, decarboxylate then catalyse the condensation between the C5 and the C2 to give pimeloyl-ACP. Please expand this.

Reply: It is an excellent suggestion. First, we have removed biotin part from Paragraph 1 of Discussion section into the first paragraph of Introduction.

Additionally, we have added the working model for BioZ action (new_Fig. S21). and, rephrased the mechanism appropriately (on P13-14, original version with changes tracked).

new_Fig.S21 A representative scheme for the sequential binding of dimeric BioZ its two fatty acyl thioesters in the ligation of glutaryl-CoA with malonyl-ACP

Q2: Fig. 3 – nice 4 structures - how were they orientated? The label is wrong. EcBioF? Please label N- and C-termini.

Reply: We appreciate the referee 2 for careful reading this manuscript and pointing out this spelling error! We have fixed this error (seen in new_Fig.4).

new_Fig.4 Structural comparison of four FAS-type enzymes

Q3: 3 BioZ homologues – show sequence alignment of these 3.

Reply: We have added sequence alignment of three BioZ homologues (**new_Fig.S5a**), arising from *A. tumefaciens*, *Brucella melitensis*, and *Mesorhizobium loti*, respectively. Among them, the similarity varies from 68.9% to 73.8%.

new_Fig.S5a Sequence alignment of three BioZ homologues

Q4: Is there an experiment to show that BioZ does catalyze the reaction between glutaryl-CoA and malonyl-CoA to give pimeloyl-CoA? It is referred to but no data.

Reply: It is an excellent question. In general, the KAS enzymes favored to exploit malonyl-ACP (rather than malonyl-CoA) as a physiological C2-donor unit to initiate the FAS II pathway. That is the reason why FabH ligates malonyl-ACP with acetyl-CoA to give C4-ACP (new_Fig.S8b). In addition to C3-ACP (new_Fig.6e), our ITC results confirmed that C5-CoA binds BioZ protein (new_Fig.8c). The assay of enzymatic activity revealed that BioZ can catalyze the ligation of glutaryl-CoA with malonyl-ACP in the context of FabG/A/I system, giving pimeloyl-ACP (new_Fig.3a).

new_Fig.S8b The *E. coli* FabH only ligates acetyl-CoA with malonyl-ACP, giving butyryl-ACP

new_Fig.3a The *A. tumefaciens* BioZ ligates glutaryl-CoA with malonyl-ACP, producing C7-ACP

new_Fig.6e ITC analysis of BioZ binding to C3-ACP

new_Fig.8c ITC analysis of interaction between BioZ and the primer glutaryl-CoA

General comments

Q1: FabB/ACP and FabZ/ACP alignment – orientate wrt catalytic triad?

Reply: It is part of **Q3** in the section of specific points. It has been addressed in **new_Fig.S20**.

new_Fig.S20 Structural insights into substrate recognition and catalysis of

BioZ and/or FabB

Q2: Could not find Table S1 – please make sure it is available.

Reply: We have fixed this spelling error, and it is Table 1. we apologize for the confusion.

Q3: Of 3 only AtBioZ is soluble. His tag on N- term then removed.

Reply: Regardless of an N-terminal 6x his tag, the BioZ of *A. tumefaciens* (AtBioZ) is exclusively soluble.

Q4: Details of EcACP, FabG, FabA and FabI, AasS, BioH?? Gel of purified proteins – Fig S7 – missing BioH? Fig S7b – no pimeloyl-ACP on gel as marker.

Reply: Thanks for referee 2's kind reminder. First, we have explained the the above nomenclature of enzymes (on P50 of change-tracking version). Namely, they refer to i) EcACP, acyl carrier protein of *E. coli*; ii) FabG, β -ketoacyl ACP reductase; iii) FabA, β -hydroxyacyl-ACP dehydratase/isomerase; iv) FabI, enoyl-ACP reductase; v) AasS, acyl-ACP synthetase of *Vibrio harveyi* B392; and vi) BioH(or BioJ), pimeloyl-ACP methyl ester carboxylesterase.

As the reviewer 2 suggested, we have added the missing BioJ enzyme (new_Fig. S7a), an equivalent to BioH. Also, a number of acyl-ACP, including pimeloyl-ACP are included in the 0.5M urea/PAGE (new_Fig. S7b).

new_Fig.S7a The enzymes of fatty acid/biotin synthesis

new_Fig. S7b A array of acyl-ACP species separated by the electrophoresis with 0.5M urea/PAGE (17.5%, pH9.5)

Q5: Fig. S8 is mass spectrometry and MS/MS analysis of ACP. This needs a Table of predicted and experimental values!!!! It is very difficult to interpret MS/MS data without a table.

Reply: It is a constructive comment that benefits readers to easily follow it. We have rephrased all the MS data, adding the calculated (predicted) and experimental value (**new_Fig.S10-S14**). In brief, as for the phosphopantetheine (Ppan)-linked pimeloyl group on ACP, the **calculated mass is 484.53** in our LC-MS/MS measurement, highly close to that of its **theoretical value, 484.1644** (**Fig. S10c**).

new_Fig.S10c LC-MS/MS determination of pimeloyl modification of ACP

Q6: Fig.S10a is the crystal structure of the AtBioZ dimer – it has electron density for glutarate and glutarate is labeled in green – was glutarate bound? There is no discussion about this – please expand!

Reply: We apologized for our improper presentation causing confusion/misunderstanding to the referee 2. We have re-organized this figure to eliminate the possible misleading information (**new_Fig.S16a-b**). First, structural architecture of the dimeric BioZ is generated in **new_Fig.S16a**. Because that we are not successful in harvesting crystal structure of BioZ and its substrate glutaryl-CoA (ACP), we performed an alternative method of molecular docking to address this question. The acyl group of glutaryl-ACP, glutarate (colored green in **new_Fig.S16b**) was subjected to molecular docking with BioZ enzyme. The rough model of BioZ-glutarate complex was created to show the approximate location of glutaryl moiety.

new_Fig.S16a-b Structural illustration of dimeric AtBioZ docked with glutarate

a. Ribbon structure of AtBioZ in dimer

b. Structure of glutarate-bound AtBioZ revealed by molecular docking

Reviewer #3

Overall comment: In this manuscript, the authors claim to have elucidated a third route to access the pimeloyl-ACP, an intermediate for the biosynthesis of biotin. Biotin biosynthesis is fragmented across bacteria, this is abundantly clear, and this manuscript attempts to answer a question that will be of broad interest to the community. This reviewer is not competent to judge the phylogenetic work and will restrict his comments to the enzymology and structural aspects of this work, in addition to the writing, all three of which are severely lacking. The following are the major deficiencies in this work:

Reply: In addition to overall summary, the referee 3 presented constructive and critical comments on this work. We appreciated these helpful criticisms. Also, we tried our best to address these questions/suggestions (point-to-point).

Q1: Major language editing is required. Several sentences, particularly in the results section make no sense, at all. The figure legends need to be descriptive. The reviewer is completely unable to understand what is being shown in Figure S8, much of the biochemical validation rests on that figure and it is impossible to understand what is implied there. If the authors haven't done so already, they are encouraged to recruit a professional manuscript preparation service. Do not insert figure callings repeatedly in the middle of sentences. Abbreviations are not to be defined in figure legends, pymol does not need to be credited in figure legends, too many other instances to parse out.

Reply: First, we would like to thank referee 3 for the suggestion of language improvement. We have revised the whole manuscript in response to all three referees' suggestion. As for figure legends, some of them are rephrased in this revision (seen in **the original version with changes tracked**). As you suggested, we have reorganized new MS/MS data in which the calculated value vs the theoretical value have been introduced (**new_Fig.S10 and S12-S14**). In fact, the referee 2 also give a similar suggestion on this figure. Respectfully, we presented an improved **Fig. S12** as an example. As for the BioZ reaction product keto-pimeloyl ACP, the he calculated mass of Ppan-linked

keto-pimeloyl moiety is 498.13, which is almost identical to its theoretical value of 498.143 (new_Fig. S12).

new_Fig. S12 LC MS/MS determination of keto-pimeloyl ACP, the BioZ reaction product

In addition, the web link of PyMol is removed. In fact, the reason we define abbreviations in figure legends is aimed to benefit readers to clearly understand them. In this revision, we have deleted unnecessary and/or redundant the explanation of abbreviations.

Q2: Biochemical validation of BioZ activity: the assertion that BioZ accepts two ACP-bound substrates needs to be validated using careful *in vitro* biochemical work. The biochemical validation, at the moment, is extremely shallow. At a minimum, the authors need to demonstrate whether or not BioZ possesses two distinct ACP binding sites or not, what is the order of substrate binding, and if the substrate binding is sequential. MS1 based assays, without MS2 fragmentation, are not conclusive. In such a rich spectra-containing innumerable ions, it is easy to "find what you are looking for" without accounting

for alternate hypotheses. Given the data presented in this study, this reviewer is not convinced that the biochemical role of BioZ is as claimed. Contemporary analytical procedures do not justify the use of data shown in Fig. 2a, without securing firm standards of all species that they claim to detect, as conclusive.

Reply: As for these comments, they are indeed constructive. Whereas, it seems likely that Reviewer 3 misunderstand the key points of BioZ action. Probably, this is due to the confused presentation in last version. Here, we would like to clearly explain it as follows:

i) The results of gel filtration and EGS-based chemical cross-linking have confirmed that BioZ is a dimeric protein (new_Fig.S5b&d). This is verified by its x-ray crystal structure. Also, it is generally consistent with all the other paradigm member of KAS enzymes, such as FabH and FabB.

new_Fig.S5b&d Biochemical evidence that BioZ is a dimer

ii) Because that binding of a certain KAS enzyme its cognate partner ACP is not strong enough to produce co-crystal of protein complex, different KAS enzymes are chemically cross-linked with ACP in the trials of complex preparation and crystal screen. Very recently, the complex structures of four KAS enzymes cross-linked with ACP (FabF-ACP, FabB-ACP, FabZ-ACP & FabA-ACP) from Burkart's research group informed us that that dimeric KAS enzyme can be bound by two ACP (one site on each monomer) ^{2,3}.

iii) Bioinformatics analyses indicated that BioZ is an evolutionarily-relevant

member within the KAS family enzyme, whose structure is determined to feature with a FabH-like dimeric configuration.

iv) Though we are not accessible to the complex structure of BioZ chemically cross-linked with ACP, we alternatively exploited ITC technique to determine the interplay between BioZ and acyl-ACP (acyl-CoA or ACP alone) (new_Fig.6e, 8c and S11). The stoichiometry of BioZ to acyl-ACP (CoA) is consistently found to be around 1:1.

v) We have provided a modified version of MS/MS for all the four C7-ACP intermediates/products, namely 5-keto-pimeloyl-ACP, 5-hydroxyl-pimeloyl-ACP, enoyl-pimeloyl-ACP, and pimeloyl-ACP (new_Fig.3b, S10, and S12-S14). As for the product, Ppan-linked 5-keto-pimeloyl-ACP from the BioZ reaction, the mass measured of 498.1533 is very close to its theoretical value of 498.1430 (new_Fig.3b). Indeed, this measurement is consistent with the migration of C7-ACP product of BioJ reaction (positive control) in conformationally-sensitive urea/PAGE.

Taken together, the integrated data allowed us to conclude the sequential binding of dimeric BioZ to its two substrates in the ligation of glutaryl-CoA (ACP) with malonyl-ACP (new_Fig.S21). This resembles other paradigm KAS member, such as FabB. In fact, similar scenarios are also seen with other two relevant enzymes BioH^{4,5} and BioW^{6,7}.

new_Fig.3b LC MS/MS analysis of the product from the BioZ reaction

new_Fig.S21 A representative scheme for the sequential binding of dimeric BioZ its two fatty acyl thioesters in the ligation of glutaryl-CoA with malonyl-ACP

Q3: BioZ structure description in the second half of Page 7 is entirely superfluous and can be omitted.

Reply: As for this point, our understanding might be different from that of Reviewer 3. This is because that BioZ structure is previously-unknown, and deserves to be described appropriately, rather than “omitted”. In the revision, we have compromised to shorten this part accordingly.

Q4: The description of the catalytic triad can be simplified, a lot.

Reply: Although the catalytic triad is very important, we have simplified it to satisfy the requirement of Referee 3.

Q5: Docking the ACP into the BioZ structure cannot be achieved using manual means only, crystal structures of both interaction partners are in hand, better computational rigor needs to be applied.

Reply: As recommended by Reviewer 3, we have tried to model BioZ-ACP complex structure using the protein-protein docking tool ClusPro developed by Kozakov *et al*⁸. We analyzed top 10 candidate models from docking results (snapshot of molecular docking-1). The location of ACP protein was closed to the binding pocket of BioZ in 8 of 10 models. However, the Ser36 residue (colored yellow) of ACP didn't orientate to the binding pocket. We speculated that these results were largely relevant to the N-terminal flexible tail of ACP. As in these models, the tail inserted into the binding pocket. In the other 2 models, ACP protein located in the dimeric interface of BioZ and thus were not considered. Next, we removed this tail of ACP to optimize docking (snapshot of molecular docking-2). Only one (No. 8) of 10 candidate models we obtained exhibits a structure that supported proposed substrate-binding model (new_Fig.S19a). Notably, it returned an implication for structure-to-function relationship. These predicted arginine residues are further consolidated by site-directed mutagenesis. Similar scenarios were seen with BioJ^{9,10} and BioH⁵.

new_Fig.S19a ClusPro-based docking of BioZ with ACP(ΔN), the truncated version of ACP without a flexible tail at N-terminus

Snapshot of molecular docking-1

Snapshot of molecular docking-2

Q6: Mutations of the basic residues can hit a lot of off-target things, again the authors are blind to alternate hypotheses.

Reply: Respectfully, we can understand the referee 3 on this issue. However, we can't be fully convinced by this comment. As for the structure-to-function study of a given enzyme, it might be well-known that the structure-guided, site-directed mutagenesis is a routine approach of gold standard. In fact, this method has been successfully applied into the other two enzymes of biotin biosynthesis. Among them, one is BioJ^{9,10}, and the other denotes BioH⁵. Thus, it is reasonable to integrate this method to test the roles of the residues in BioZ function.

Q7: At the very least, the use of biophysical techniques such as ITC and/or SPR is essential to map out the ACP:BioZ binding stoichiometries.

Reply: We do agree with this comment raised by Reviewer 3. In fact, the referee 1 also recommended us to conduct ITC. We carried out ITC experiments to address the binding of BioZ to glutaryl-ACP, malonyl-ACP, and ACP alone (new_Fig.6e, S11 and S19b). As expected, the stoichiometry of BioZ to its partner is around 1:1.

new_Fig.6e ITC analysis of BioZ binding to C3-ACP

Q8: Manual modeling followed by *in vivo* testing does not justify the hypothesis here.

Reply: It is identical to **Q5**. First of all, I have to declare that it is molecular docking, rather than manual manipulation. Second, a putative interface between BioZ and ACP is proposed, on which four arginine residues are highlighted. Third, these arginine residues are further consolidated by site-directed mutagenesis. Similar scenarios were seen with BioJ^{9,10} and BioH⁵. To the best of our knowledge on biotin synthesis thus far, we believe the interpretation is reasonable and reliable.

Q9: Again, the reviewer does not find justification that two ACPs can bind to BioZ, or, that ACP binding can be sequential given the structural data presented. This is the crux of the biochemical activity, one ACP binds, transfers the payload to the active site, and then the second ACP binds. The structural data needs to support this hypothesis. At the moment, it does not.

Reply: In principle, the comment raised by the reviewer 3 is not in problem. However, it might be not technically feasible right now. This is mainly due to weak binding of a given KAS enzyme to a universal partner ACP. That is why Burkart's research group employed the method of chemically cross-linking to produce the complexes of four KAS enzymes (FabF-ACP, FabB-ACP, FabZ-ACP & FabA-ACP) for crystallography. Indeed, the resultant structures of KAS-ACP informed us that the dimeric KAS enzyme bind two ACP (one site on each monomer)^{2,3}. Even though we cannot capture an intermediate of BioZ-ACP, our ITC result revealed that the stoichiometry of BioZ to ACP is 1:1. Thus, it is reasonable to propose a working model of sequential binding of BioZ to its substrate acyl-ACP (CoA) (new_Fig. S21). Practically, this is the best we can do at present, and supposed to meet the requirement by our journal. However, this requires further demonstration in the future.

new_Fig.S21 A representative scheme for the sequential binding of dimeric BioZ its two fatty acyl thioesters in the ligation of glutaryl-CoA with malonyl-ACP

Related references

1. Sullivan, J.T., Brown, S.D., Yocum, R.R. & Ronson, C.W. The *bio* operon on the acquired symbiosis island of *Mesorhizobium* sp. strain R7A includes a novel gene involved in pimeloyl-CoA synthesis. *Microbiology* **147**, 1315–22 (2001).
2. Milligan, J.C. et al. Molecular basis for interactions between an acyl carrier protein and a ketosynthase. *Nat Chem Biol* **15**, 669–671 (2019).
3. Mindrebo, J.T. et al. Gating mechanism of elongating beta-ketoacyl-ACP synthases. *Nat Commun* **11**, 1727 (2020).
4. Lin, S., Hanson, R.E. & Cronan, J.E. Biotin synthesis begins by hijacking the fatty acid synthetic pathway. *Nat Chem Biol* **6**, 682–8 (2010).
5. Agarwal, V., Lin, S., Lukk, T., Nair, S.K. & Cronan, J.E. Structure of the enzyme-acyl carrier protein (ACP) substrate gatekeeper complex required for biotin synthesis. *Proc Natl Acad Sci U S A* **109**, 17406–11 (2012).
6. Wang, M. et al. Using the pimeloyl-CoA synthetase adenylation fold to synthesize fatty acid thioesters. *Nat Chem Biol* **13**, 660–667 (2017).
7. Estrada, P. et al. The pimeloyl-CoA synthetase BioW defines a new fold for adenylate-forming enzymes. *Nat Chem Biol* **13**, 668–674 (2017).
8. Kozakov, D. et al. The ClusPro web server for protein-protein docking. *Nat Protoc* **12**, 255–278 (2017).
9. Wei, W. et al. Molecular basis of BioJ, a unique gatekeeper in bacterial biotin synthesis. *iScience*, DOI:<https://doi.org/10.1016/j.isci.2019.08.028> (2019).
10. Feng, Y. et al. A *Francisella* virulence factor catalyses an essential reaction of biotin synthesis. *Mol Microbiol* **91**, 300–314 (2014).

Reviewers' comments:

Reviewer #1 (Remarks to the Author):

Most concerns have been satisfactorily addressed. The conclusions are significantly strengthened with the additional experimental results. A few minor points: (1) significant figures-for example-don't believe it is as accurate as $K_d = 3.250 \pm 0.636 \mu\text{M}$ (Figure 8); (2) Fig. S13--it is not clear how the Ppan-linked 5-hydroxyl-pimeloyl moiety was calculated to be 500.147, it was probably figured out from the difference between y16 and y17, please make this clear. There are two peaks labeled as y16+-H₂O-----this is an apparent error, please double check the assignments and labels of all other peaks; (3) Fig. S10 and S14 have similar problems as Fig. S13.

Reviewer #2 (Remarks to the Author):

Many thanks for taking time to make the major edits suggested by each of the 3 referees. You can respond very well with new biochemical data (MS, assays, molecular docking) and re-worked many of the figures. They are now very good and the data you include supports your conclusions. I am also pleased that you got help in writing the scientific English to greatly improve the manuscript. Your BioZ work will be a valuable addition to the field.

I confirm that I think your paper should now be accepted.

Also, you should note and cite a paper on BioZ by the Cronan group that was also published in Nature Comms while you were revising your manuscript - it does not have the 3d structural/x-ray work that your paper has but together they are a nice pair.

<https://www.nature.com/articles/s41467-020-19251-5>

Response letter

Reviewer #1

Overall comment: Most concerns have been satisfactorily addressed. The conclusions are significantly strengthened with the additional experimental results.

Reply: We appreciate referee 1 for the positive evaluation on this revision.

A few minor points:

Q1: significant figures-for example-don't believe it is as accurate as $K_d = 3.250 \pm 0.636 \mu\text{M}$ (Figure 8)

Reply: First of all, we would like to thank the reviewer 1 for the careful reading. Second, we should apologize for our inappropriate (unclear) presentation of ITC data, which might cause the referee 1's misunderstanding to some extent. In fact, we carried out three independent ITC experiments (accessible to source data), and a representative graph is given in the manuscript. Thus, resultant stoichiometry values (N and K_d) from three independent experiments are given in an average \pm SD. Because of the referee 1' concern with ITC data precision, we re-analyzed and processed the raw data with an updated version **Microcal PEAQ-iTC system rather than the routine iTC200 system (last generation)**. The resultant data is given (**new_fig.8c**). Of note, the K_d value has been updated from $3.250 \pm 0.636 \mu\text{M}$ to **$2.64 \pm 0.23 \mu\text{M}$** . In addition, we also rephrased the figure legend of Fig. 8 accordingly (**on L7-8, P30 & L7-8, P31 of change-tracking version**).

new_fig.8c ITC analysis of BioZ binding glutaryl-CoA

Q2: Fig.S13--it is not clear how the Ppan-linked 5-hydroxyl-pimeloyl moiety was calculated to be 500.147, it was probably figured out from the difference between y16 and y17, please make this clear. There are two peaks labeled as y16+-H₂O-----this is an apparent error, please double check the assignments and labels of all other peaks; (3) Fig. S10 and S14 have similar problems as Fig. S13.

Reply: It is a good comment. As the reviewer 1 suggested, we have modified it accordingly (**new_fig. S13**). As for the two peaks “y16+-H₂O”, we have double-checked the raw MS/MS data. We appreciated the referee 1’s careful reading, and eliminated this error introduced by mistake in the graph editing/organization. Here, we presented a correct version (**new_fig. S13**). Similarly, we also checked two additional figures (**new_fig. S10** and **new_fig. S14**) and highlighted the peaks for mass calculation with **pink arrows**.

new_fig. S13 MS/MS analysis of 5-hydroxyl-pimeloyl ACP intermediate from the *in vitro* BioZ reaction

Reviewer #2

Overall comment: Many thanks for taking time to make the major edits suggested by each of the 3 referees. You can respond very well with new biochemical data (MS, assays, molecular docking) and re-worked many of the figures. They are now very good and the data you include supports your conclusions. I am also pleased that you got help in writing the scientific English. It greatly improves the manuscript. Your BioZ work will be a valuable addition to the field. I confirm that I think your paper should now be accepted.

Reply: We do appreciate the reviewer 2 for the highly-positive recognition of current version of this BioZ work with nice and sound data. Obviously, referee 2 gives a comprehensive & meaningful summary on the impact of the major findings in the related field.

Q1: Also, you should note and cite a paper on BioZ by the Cronan group that

was also published in Nature Comms while you were revising your manuscript - it does not have the 3d structural/x-ray work that your paper has but together they are a nice pair. <https://www.nature.com/articles/s41467-020-19251-5>

Reply: This is a subjective comment. As the referee 2 mentioned, we have already discussed the finding of Cronan group and cited the literature in the revision (on L23-28, P14). The discussion part appears as follows “ Of note, during the revision of this manuscript, Hu and Cronan reported a genetic and enzymatic study of BioZ action, pointing out lysine catabolism as the source of glutaryl-CoA, an intracellular priming substrate for BioZ¹. In generally consistent with the description of Hu and Cronan¹, we presented integrative evidence *in vivo* and *in vitro* (covering bacterial genetics, biochemistry and chemical biology) for BioZ activity”.

In addition, the reference is numbered 47 in the section of reference [Hu, Y. & Cronan, J.E. alpha-proteobacteria synthesize biotin precursor pimeloyl-ACP using BioZ 3-ketoacyl-ACP synthase and lysine catabolism. *Nat Commun* **11**, 5598 (2020)].

Related references

1. Hu, Y. & Cronan, J.E. alpha-proteobacteria synthesize biotin precursor pimeloyl-ACP using BioZ 3-ketoacyl-ACP synthase and lysine catabolism. *Nat Commun* **11**, 5598 (2020).